



# The application of new distribution in determining extreme hydrologic events such as floods

Łukasz Gruss[1], Jaroslav Pollert (Jr.)[2], Jaroslav Pollert (Sr.)[2], Mirosław Wiatkowski[1], Stanisław Czaban[1]

[1]Institute of Environmental Engineering, Wrocław University of Environmental and Life Sciences, Wrocław, 50-375, Poland
[2]Faculty of Civil Engineering, Czech Technical University in Prague, Praha, 166 29, Czech Republic

*Correspondence to*: Łukasz Gruss (lukasz.gruss@upwr.edu.pl)

**Abstract.** In hydrology, statistics of extremes play an important role in the use of time series analysis as well as in planning, design and operation of hydrotechnical structures and water systems. In particular, probability distributions are used to estimate and forecast floods. However, in order to use distributions, the data must be random, with a change-point and should not have a trend. Unfortunately, the data being analyzed are not independent, which is very often due to the anthropogenic impact, among other factors. In situations where various processes generate rainfall and floods in river basins, the use of mixed distributions is recommended. However, an accurate estimation of multiple parameters derived from a mixture of distributions can be difficult, which is the biggest disadvantage of this approach. Therefore, as an alternative, we propose a new distribution – the Dual Gamma Generalized Extreme Value Distribution (GGEV) developed by Nascimento, Bourguignony and Leão (2016). We compared this distribution with selected 3-parameter distributions: Pearson type III, Log-Normal, Weibull and Generalized Extreme Value. In addition, various methods of estimating 3-parameter distributions were used. As a case study, rivers from Poland and the Czech Republic were investigated, because this has a significant impact on water management in the Upper Oder basin due to the strategic water reservoirs and other hydrotechnical constructions, either existing or planned. Currently, there are no clearly indicated distributions for the Upper Oder basin. Therefore, our aim was to approximate them. Two methods were used, namely the Annual Maximum (AM) and the Peaks Over Threshold (POT). In the latter case, two methods for determining the threshold were used, namely: the Mean of the Annual Maximum River Flows (MAMRF) and the Hill plot. Hence, the basic 3-parameter Weibull distribution, with parameters estimated using the modified method of moments and the maximum likelihood estimation, yielded a better fit to the observation series in the AM and POT methods. For the AM and POT (MAMRF, Hill plot) methods, the GGEV turned out to be the best-fitted distribution according to the Mean Absolute Relative Error (MARE). The GGEV distribution can be used as an alternative to mixed distributions in various samples, both homogeneous and heterogeneous. This distribution turned out to be the best fit especially for the sample whose independence is affected by the presence of a GGEV water reservoir.





# 1 Introduction

Natural disasters, especially fluvial floods, are a serious natural hazard in western and central Europe (Gvoždíková and Müller, 2017; Kundzewicz, et al. 2005). In this part of Europe flash floods and river floods occur (Gvoždíková and Müller, 2017). Higher and more intense rainfall may increase the frequency of extreme floods (Barredo, 2007; Christensen and Christensen, 2003; Pollert, 2006).

       Many floods of different intensity and extent took place on the Oder and its tributaries in the 20th century and in the
beginning of the 21st century (Dubicki et al., 2005). The flood that occurred in Poland in the Oder and the Vistula basins in the summer 1997 caused 54 fatalities and material losses estimated at billions of USD (Kundzewicz et al., 1999). Extreme fluvial flooding took place in many parts of the Czech Republic in August 2002. This flood overwhelmed most of existing flood protection systems and caused damage exceeding EUR 3 billion (Holický and Sýkora, 2010). During the catastrophic flood in the Otava river basin in August 2002 critical structures (such as: railway embankments, undersized bridges or
culverts) were located in the upper part of the river basin, where they affected the forming flood wave, as well as in the lowland agricultural regions (Langhammer and Vilímek, 2008). Furthermore, all floating objects from the river must be moved to harbours to prevent any damage they might cause if carried away by the stream during the flood (Buchlák et al., 2019). As a result of heavy rain events, an extreme hydrological situation occurs in a water body, which is manifested by surface runoff from the urbanized catchment in excess of the computational drainage capacity of the urbanized area (Pollert,
45    2006).

       The maximum flows observed during floods over the years constitute the basis for the calculation of exceedance probabilities (Szulczewski and Jakubowski, 2018). An appropriate estimation of flow values for various probabilities of occurrence or various return periods, which are referred to as design floods, is of fundamental importance for the management of water resources (Alila and Mtiraoui, 2002). Because these values are of great significance in planning,
design and operation of hydrotechnical structures and water systems, hydrologists take frequent recourse to time series analysis (Mamman et al., 2017). According to Ahaneku and Otache (2014) and Mamman et al. (2017), time series are used for the development of mathematical models, which are made to generate synthetic hydrologic records, forecast hydrologic events, detect intrinsic stochastic characteristics of hydrologic variables and extend records or fill those that are missing. The statistics of extremes have played an important role both in the design of hydrotechnical structures and in water management
(Cassalho et al., 2018; Dunne and Leopold, 1978; Katz et al., 2002; Młyński et al., 2018; Szulczewski and Jakubowski, 2018). Probability distributions are used especially for flood estimation and prediction in reservoir inflow analysis, and at any hydropower dam (Dunne and Leopold, 1978; Mamman et al., 2017), flood embankments and bridges (Dunne and Leopold, 1978; Mamman et al., 2017; Bao et al. 1987), or culverts (Bao et al., 1987), where the accuracy of these methods has a profound significance for economic investments (Bao et al., 1987; Mamman et al., 2017). Therefore, we decided that
our research would be carried out on rivers, whose proper use has a significant impact on water management of the Upper Oder basin. On such rivers, the Mała Panew or the Widawa, there are multi-purpose water reservoirs. The Turawa reservoir



located on the Mała Panew has several functions, but its primary role is to store water from the Mała Panew for navigation, power generation, fishing and recreation, and to ensure protection against floods (Wiatkowski and Wiatkowska, 2019). Functions of the Michalice reservoir on the Widawa river include water storage, flood protection, electric energy generation,

fishing (non-industrial pisciculture) and other recreational purposes as well as agricultural irrigation (Gruss et al., 2019). On rivers such as the Budkowiczanka or the Biała Głuchołaska, hydrotechnical facilities could be built to store water.

New water reservoirs might be built to increase the water resources in the Upper Oder basin. However, this requires that a hydrological analysis of observed data is carried out.

       Long time observation series were processed using the Flood Frequency Analysis (FFA), so that the distribution

analyzes could be carried out later. FFA is often adopted to investigate the relationships between flood magnitude and the corresponding frequency of occurrence (Gharib et al., 2017). FFA is also used to fit a probability distribution to an empirical distribution function (Rahman et al., 2015; Haktanir, 1991; Lang et al., 1999; Silva et al., 2012; Yadav and Pande, 1998). In sampling of extreme flood values from observed flow series, two approaches are the most common: Annual Maximum (AM) and Peaks Over Threshold (POT) (Bezak et al., 2014; Gharib et al., 2017; Langbein, 1949; Lang et al., 1999; Madsen et al.,

1997; Svensson et al., 2005; Wang, 1991). In time series modelling, sample independence is very important. According to Alexandersson (1986), this is related to the access to reliable data, free from artificial trends and changes. However, Rutkowska (2015) points out that the existence of a trend in a hydrological sequence of observations is a sign of heterogeneity as a result of climate change or anthropogenic activity. Also Cassalho et al. (2018) indicate that stationarity of the hydrological records may contain heterogeneities due to anthropogenic actions. In turn, Barets (1982) reports that testing

of sample randomness is of fundamental importance in statistics.

       Over the past 20 years, research has been conducted on testing various distributions and methods for estimating their parameters have been developed. According to Holický and Sýkora (2010), for the annual maximum flows of the Vltava River in Prague (Czech Republik), 2-parameter distributions such as Pearson type III (2P3) and Log-Normal (2LN) seem suitable. In Poland, in the Upper Oder basin, the tested distributions are: Pearson type III (P3), Log-Normal (LN) and

Generalized Extreme Value (GEV), mixture of gamma and GEV (Mix Gam+GEV) (Szulczewski and Jakubowski, 2018) and 3-parameter distributions such as Pearson type III (3P3) and Log-Normal (3LN) and Weibull (3W) (Gruss et al., 2019). Moreover, Młyński et al. (2019), who analyzed the tributaries of the Upper Vistula and studied 2-parameter distributions such as 2P3, Weibull's (2W) and 2LN, have found that the 2LN was the best fitted. On the other hand, Bezak et al. (2014) investigated the Litija river in Slovenia and considered distributions such as LN, P3, log-Pearson type III (LP3), Gumbel

(Gum), GEV and generalized logistic (GL). For these distributions they propose to use the method of L-moments to estimate the distribution parameters, and the best-fitted distributions are: LP3, GEV, P3 and GL. However, the LP3 distribution is mostly used in Slovenia. Bačová-Mitková and Onderka (2010) report that for the Danube river, in Slovakia the LN and LP3 distributions are most commonly used. In addition, as stated by Stojković et al. (2017), in Serbia and in the USA, the recommended distribution is LP3, whereas in the United Kingdom, the GEV distribution is recommended by the Natural

Environment Research Council and the maximum likelihood estimation method is used for the assessment of the GEV





parameters. In turn, as reported by Gruss et al. (2018) in the United Kingdom, Reed and Robson (1999) recommend using the GL distribution. Further, according to Gamage et al., (2013) in Australia the 2P3 distribution was successfully fitted to the data observed. Northern Tunisia the Generalized Normal (GNO) distribution is used, while in central/southern Tunisia the best fit for the data observed was achieved using the GNO and GEV (Abida and Ellouze, 2008).

Beskow et al. (2015) and Cassalho et al. (2018) report that in Brazil the LN2, LN3 and Gum are the most commonly used. Moreover, Cassalho et al. (2018) reported that the performance of multiparameter distributions such as Kappa (KAP) and Wakeby (WAK) in statistical modelling of the observed maximum annual streamflow series in Brazil was better than that of traditional 2-parameter distributions. Haktanir (1991) studied many various distributions, out of which the best fit was achieved using the LN2 and the Gum. Cunnane (1979) analyzed the negative binomial distribution, which was worse than

the Poisson distribution for data from the 26 measuring stations in Great Britain. In turn, Lang et al. (1999) analyzed the Exponential distribution among others. As reported by Kidson and Richards (2005), in 2-parameter distributions estimators are easier and faster to fit; however, 3-parameter distributions have a third additional parameter – the scale, which ensures more flexibility and allows fitting a larger number of catchment records. The 3-parameter models are rarely used, which is precisely why we decided to use them in our research.

Various methods of estimating distribution parameters have been studied. Different scientists came to different conclusions. For the LN, P3 and GEV the Maximum Likelihood Estimator (MLE) is recommended (Szulczewski and Jakubowski, 2018), whereas the L-moments method was used for the GEV, LN3, P3, GLO, KAP and WAK (Cassalho et al., 2018). The Method of Moments (MM) and the MLE (among others) were used for the 3LN, 3P3, 3W (Gruss et al., 2019), LN, P3 and LP3 (Haktanir, 1991), and the MLE for the 2LN, LP3, GEV and 2W (Kidson and Richards, 2005). As reported

by Cohen and Whitten (1980), the Modified Method of Moments Estimation (MMM) can be used for distributions such as the 3LN. In the case of the GEV distribution, Hosking et al. (1985) recommend using the probability weighted moments (PWM) method, while Smith (1985) studied the MLE method. Madsen et al. (1997) proved that estimation methods such as the MM and PWM in the GPA distribution in the POT method are preferred for negative shape parameters (for heavy-tailed distributions), while the GEV distribution in the AM method provides the most efficient estimator for positive shape

parameters. They recommend using the POT method for estimating parameters using the MM: (1) for negative shape parameters, (2) with exponentially distributed exceedances, if the shape parameter is close to zero. In the AM method, however, using the MM estimation for moderately positive shape parameters, and for the MLE estimation for large positive shape parameters is recommended. This is confirmed by Gharib et al. (2017), who used various methods of estimating the Generalized Pareto Distribution (GPD) and eventually concluded that, for short tails, the MLE is better.

Factors such as anthropogenic impact, climate change or spatial distribution of precipitation generate changes in the frequency of observed floods. Consequently, high peaks, trends or long-term periodic oscillations in flood records appear (Stojković et al., 2017). Assuming, that the pattern of rainfall and flood generation in river catchments can be different, the maximum annual flood time series can consist of various processes taking place in the catchment. For this reason, the predefined distribution functions are not always the best fitted ones. Besides, there is a problem of genetic heterogeneity.





This means that the best fit to an empirical distribution of two or more processes is achieved by using a mixture of two or more distributions (Hess et al., 2005; Stojković et al., 2017; Szulczewski and Jakubowski, 2018).

Some researchers analyzed mixed distribution. Stojković et al., (2017) investigated long series of observations of the maximum annual flows from the Kolubara river. Their research shows, that the best fit to the empirical distribution function of the flood peaks is provided by the mixed LP3, mixed P3 and mixed GEV. One of advantages of these distribution

functions is that they can be adapted to empirical data with considerable skewness, which is quite pronounced in the case of flood peaks. Vaidyanathan and Lakshmi (2016) propose a multivariate gamma mixture model (MGMM) with independent marginals. In order to estimate the parameters of this distribution, they use the Wilson-Hilferty approximation, the MCLUST algorithm and the principle of maximum likelihood. Escalante-Sandoval (2007) studied a mixed distribution with EV1 and GEV as components and analyzed heterogeneous samples from 35 gauging stations from North western Mexico. The

maximum likelihood method was used to estimate six parameters of this model. In turn, Szulczewski and Jakubowski (2018) compared the basic distributions: P3, GEV, LN with a mixed distribution, which is a Mix Gam+GEV. In order to estimate this six-parameter distribution, they used the maximum likelihood method and a genetic algorithm. This distribution provided the best fit, in terms of the Mean Absolute Relative Error (MARE), to the samples from the Upper Oder Basin that are heterogeneous.

Nascimento et al. (2016) presented three new distributions: The Dual Gamma Generalized Extreme Value Distribution (GGEV), the Exponentiated Generalized Extreme Value Distribution (EGEV), the Transmuted Generalized Extreme Value Distribution (TGEV). They have an additional skewness parameter and introduce this varying tail weight. This makes them more flexible than the GEV. The parameter estimation of these new distributions is done under the Markov chain Monte Carlo (MCMC) approach. Their team conducted tests on two real data sets. The first set consisted of the monthly maxima of

water levels for the Gurgueia River, located in the State of Piauí, Brazil, in 1975-2012. The second consisted of the maximum precipitation in 1931-2008 for the Barcelos Station in the north of Portugal. In both applications the GGEV model provided definitely the best fit out of the three.

        The aim of the study was to compare 3-parameter distributions (P3, LN3, W3, GEV) with the new distribution – GGEV proposed by Nascimento, Bourguignony and Leão (2016). Studies were carried out for both independent and non

independent samples in the POT and AM methods. As a case study, rivers from Poland and the Czech Republic from the Upper Oder basin were used.

Currently, there are no clearly indicated distributions for the Upper Oder basin from Poland and the Czech Republic, hence we found it interesting to approximate them.

In our study we hypothesized that the GGEV distribution is the best-fitted distribution for samples, in the Upper Oder basin,

for which the flow phenomenon was caused by anthropogenic activity in the catchment. Additionally the GGEV distribution is the best suited empirical distribution irrespective of sample independence.



## 2 Materials and Methods

### 2.1 Study area

Six tributary profiles of the Oder, located in the Upper Oder Basin, on the territory of Poland and the Czech Republic were
used for the analysis. There are three lowland rivers: the Budkowiczanka, the Mała Panew (2 profiles) and the Widawa, as
well as two mountain rivers: the Biała Głuchołaska and the Osobłoga (fig. 1). The catchments of these last two rivers are

The Budkowiczanka River is 56.5 km long. This river flows into the Stobrawa River, a right-hand tributary of the Oder (Fig.
1). The river flows west and has an average slope of 1.825‰. The width of the river is 3-8m. The analyzed water gauge
profile is located at km 18.43 of its course.

The second analyzed river is the Mała Panew, which is 129.1 km long. It is a right-hand tributary of the Oder. Its average
slope is 1.58‰. The Turawa Reservoir is located at 18.9 km of its course. The capacity at Normal Pool Level is 80.04 MM
$m^3$ (Wiatkowski and Wiatkowska, 2019). One of the gauge stations (Turawa profile) is below this reservoir. The distance
from it to the reservoir dam is 1.57 km. Another water gauge (Staniszcze Wielkie profile) is located above the reservoir, and
the distance from it to the reservoir is about 13.9 km.

The Widawa is 114.6 km long and flows into the Oder. The average slope of its channel is 1‰. The Michalice reservoir is
located on the Widawa, at km 70.232 of its course. Its capacity at Normal Pool Level is 1.19 MM $m^3$ (Gruss et al., 2019).
The distance between the water gauge in the Zbytowa profile and the dam of the Michalice reservoir is 27.5km.

The Osobłoga is a 65.5 km long mountain river. It is a left-hand tributary of the Oder. Its average slope ranges from 1.3 to
3.4‰. A dry dam is planned in the valley of the Osobłoga at km 26.4 of its course with a Normal Pool Capacity of 1 MM
$cm^3$.

The Biała Głuchołaska (Czech Bělá) is a 54.9 km long mountain river. It flows into the Nysa Kłodzka (a left tributary of the
Oder). The average channel slope of the transgenic Biała Głuchołaska is 1.87‰ and 11.22‰ in Poland and in the Czech
Republic, respectively.

Table 1 presents the characteristics of the analyzed profiles.


### 2.2 Methods

The AM (Haktanir, 1991; Yadav, 1998) and POT methods were used for analyzing extreme hydrologic events based on long
time series data (Bezak et al., 2014; Gharib et al., 2017; Katz et al., 2002; Kidson and Richards, 2005 et al., Kundzewicz et
al., 2005; Madsen et al., 1997; Svensson et al., 2005). According to Bačová-Mitková and Onderka (2010), Bezak et al.
(2014), Gharib et al. (2017), Langbein (1949), Lang et al. (1999), Kundzewicz et al. (2005), Svensson et al. (2005) the AM
method is the most common because it samples only one extreme event per year. The POT includes all peaks above a certain
flow value (the threshold) (Bezak et al., 2014; Gharib et al., 2017; Kundzewicz et al., 2005; Svensson et al., 2005). Daily



water flows and maximum monthly flows for the analyzed streams come from the Institute of Meteorology and Water Management - National Research Institute in Warsaw, Poland.

In order to simplify notation, the following abbreviations for rivers and profiles will be used: the Budkowiczanka (the Krzywa Góra profile): Bu, the Biała Głuchołaska (the Głuchołazy profile): BB, the Mała Panew (the Staniszcze Wielkie profile): MPSW, the Mała Panew (the Turawa profile): MPT, the Osobłoga (the Racławice Śląskie profile): O, the Widawa (the Zbytowa profile): Wi.

Abbreviations used in different parts of the article are included in Appendix A.

**2.2.1 Homogeneous tests**

Long time series of the six profiles were checked for trend, randomness and change point detection.

The Mann-Kendall test (MK) is frequently used to detect monotonic trend in long time series of hydrological data (Cassalho et al., 2018; Rutkowska, 2015; Svensson et al., 2005). This nonparametric test is used to check if data is identically distributed (Libiseller et al., 2002; Mann, 1945; Kendall and Gibbons, 1990). This test was successfully used by

Svensson et al. (2005), who studied trend detection in river flow series at 21 stations worldwide or by Cassalho et al. (2018), who assessed the trend of the rivers in the state of Rio Grande do Sul in Brazil, and also by Młyński et al. (2018). Rutkowska (2015) reported that the Mann-Kendall test for long series is stronger than the Cox-Stuart test.

A two sided test was performed. The null hypothesis is that the data are identically distributed, the alternative hypothesis is that the data follow a monotonic trend. In nonparametric tests: MK (Bezak et al., 2014; Cassalho et al., 2018),

the significance level was set at 5%.

The Standard Normal Homogeneity Test (SNHT) (Alexandersson, 1986) is used to analyze the change-point. In this test we followed the methodology and used the critical value provided by Khaliq and Ouarda (2007). The test was used to detect a inhomogeneities and undocumented discontinuities in hydrological series with continuous data. The null hypothesis was that one or more distributions had the same location parameter (no change), the alternative hypothesis was that there was

a change point.

In the Bartels test for randomness (Bartels, 1982), (B) the null hypothesis that the sample is random is tested against the alternative hypothesis that the data is significantly different from random. A two sided test was performed.

The Non-Parametric Trend Tests (MK, B) and Change-Point Detection (SNHT) were implemented using the R Package 'trend'.

**2.2.2 AM model**

The AM series approach is the most used method in FFA in many countries. Based on the values of the maximum monthly flows of the analyzed river profiles, the maximum annual values of flows from a multiyear period were selected in samples.



### 2.2.3 POT model

Before selecting the optimal threshold, the analyzed daily series were verified using the two independence criteria described by Cunnicane (1979) and applied by Bačová-Mitková and Onderka (2010), Bezak (2014), Madsen et al. (1997), Silva et al. (2012). These criteria are as follows: 1) consecutive peaks must be separated by 3Tp, where Tp is the average time to peak of the first five clean hydrographs on the record, 2) the smallest flow value between two consecutive peaks must be higher than two-thirds of the first peak value of the wave.

230        The optimal threshold should be selected. It cannot be too high because the variance will increase by reducing the number of events, nor too low due to maintaining the assumption of independent and identically distributed flood variables (Gharib et al., 2017). Therefore, for the optimal threshold detection, we chose the graphic method called the Hill plot and the analytical method called the Mean of the Annual Maximum River Flows (MAMRF). The MAMRF is described by Kundzewicz et al. (2005) and it was used to verify the results of Hill plot. The application of the POT MAMRF method for

the selection of threshold is shown in fig. 3.

The MAMRF threshold was set by collecting the maximum annual flows from the analyzed multiyear period. The average value was determined for these flows.

According to Gharib et al. (2017) graphic methods can be easily applied. However, the disadvantage of these methods is that the interpretation of the outcome may be vague and that it is sometimes difficult to determine which part of the chart is

completely linear. Despite that, the proposed Hill plot method was successfully used by Andreeva et al. (2012) in relation to the study of the distribution of financial gains/losses. According to researchers, Hill plot is a good instrument to find the optimal threshold. The threshold was found in line with the guidelines given by these researchers.

In our opinion, the MAMRF method can be used for the series in Central Europe and should be tested for other parts of the world.

### 2.2.4 Probability distributions for the POT and AM modelling

The same distributions were used in both methods: 3LN, Pearson Type III (3P3), GEV, Weibull (3W). All these are three-parameter distributions. Moreover, the authors verified a four-parameter distribution called Dual Gamma Generalized Extreme Value Distribution (GGEV) described by Nascimento et al. (2016).

The probability density function (PDF) of the 3P3 distribution is given by:

$$f(x) = \frac{1}{|s|^\alpha \Gamma(\alpha)} |x - \lambda|^{\alpha-1} e^{-\frac{x-\lambda}{s}},$$   (1)

for s≠0, a>0 and $\frac{x-\lambda}{s} \geq 0$.

Where:

α, s, λ are shape, scale and location parameters, respectively.





The MM and the MLE were used to estimate the parameters for the 3P3 distribution. The Method of Moments is based on

the empirical input moments such as: mean, variance, skewness and kurtosis of the sample data. In the MLE, the idea is to

determine those parameter values for which the logarithm of the likelihood function is maximal. The likelihood function is

proportional to the probabilities of occurrence of all the individual elements in the sample. The probability of this sample

must be maximal, because the sample observed comes from many other possible samples (Haktanir, 2009).

In the gamma distribution developed by Becker and Klößner (2017), this function allows negative scale parameters to allow

for negative skewness. The estimation of the parameters and fitting of probability distribution was done using the R package

'PearsonDS'.

The multiparameter probability distribution function of the 3LN described by Alila and Mtiraoui (2002), Cassalho et al.

(2018), was obtained by the R package 'EnvStats' and is given by the formula:

$$f(x) = \frac{1}{(x-\alpha)\sigma_y\sqrt{2\pi}} \exp\left\{-\frac{1}{2\sigma_y^2}[\log(x-\alpha) - \mu_y]^2\right\}, \tag{2}$$

where:

$\mu_y$, $\sigma_y^2$, $\alpha$ are location, scale and shape parameters, respectively.

The 3LN is similar to the 2LN, except that x is subtracted by a value $\alpha$ in the former, which represents the lower bound

(Cassalho et al., 2018). The Parameters of this distribution were estimated by: MM described by Johnson et al. (1994),

MMM given by Cohen (1988) and MLE as shown by Meeker and Escobar (1998). The estimation of the parameters and

fitting of probability distribution was done using the following R packages: 'EnvStats' and 'weibulltools'. For all the methods

used to estimate the distribution parameters a confidence level of 0.95 was assumed.

Three-Parameter Weibull Distribution PDF expressed by equation 3 is described by Teimouri and Gupta (2013)

$$f(x) = \frac{\alpha}{\beta}\left(\frac{x-\mu}{\beta}\right)^{\alpha-1} e^{-\left(\frac{x-\mu}{\beta}\right)^\alpha}, \tag{3}$$

for x>μ and α and β are greater than 0. The parameters α, β and μ are the shape, scale and location parameters, respectively.

As in the case of 3LN, for this 3W distribution, the estimation of the three function parameters was carried out using the

MM, MMM and MLE. The following R packages have been used to estimate the parameters of the probability distribution:

'weibulltools','Fadist', 'ForestFit'.

The Generalized Extreme Value Distribution (GEV) was used in the research of Abida and Ellouze (2008), Bezak,

Brilly and Šraj (2014), Cassalho et al. (2018), Kidson and Richards (2005), Szulczewski and Jakubowski (2018). The PDF

function is given in equation 4:

$$f(x) = \exp\left[-\left\{1 + \frac{s(x-\alpha)}{b}\right\}^{-1/s}\right], \tag{4}$$

where:

α, b, s are location, scale and shape parameters, respectively.





for 1+s(x-α)/b>0, where b>0.

The parameters of this distribution were estimated by: MLE, as described by Smith (1985) and PWM, as in Hosking, Wallis and Wood (1985). The estimation of the parameters and fitting of GEV distribution was done using the following R packages: 'evd' and 'fExtremes'.

The GGEV probability density function proposed by Nascimento et al. (2016), is given by formula:

$$f(x;\ \mu;\ \sigma;\ \xi;\ \delta) = \begin{cases} \frac{\sigma^{-1}}{\Gamma(\delta)}[1 + \frac{\xi(x-\mu)}{\sigma}]^{-(\frac{\delta}{\xi})-1} \exp\left\{-[1 + \frac{\xi(x-\mu)}{\sigma}]^{-\frac{1}{\xi}}\right\}, \xi \neq 0 \\ \frac{\sigma^{-1}}{\Gamma(\delta)}exp\{-\delta[(x-\mu)/\sigma]\}exp\left\{-exp\left\{[-\frac{x-\mu}{\sigma}]\right\}, \xi \to 0 \end{cases},$$ (5)

where:

μ – location parameter

σ – scale parameter

ξ – shape parameter

δ – is shape parameter of GGEV extension.

A Bayesian Monte Carlo Markov Chain (MCMC) approach is used to estimate the posterior parameters of the GGEV distribution (Nascimento et al. 2016).

The estimation of the parameters and fitting of probability distribution was done using the following R packages: 'MCMC4Extremes'.

### 2.2.5 Goodness-of-fit tests

One of the goals of this article was to propose a new GGEV distribution model in the AM and POT method. For this reason, we checked whether this distribution or the 3-parameter distributions used in these studies provided the best fit to the empirical distribution function. The Chi-squared Test ($\chi^2$), Kolmogorov-Smirnov (K-S), and the Mean absolute relative error (MARE) tests were widely used to indicate the adequacy of the distribution functions being tested.

The $\chi^2$ test was used to compare the selected distribution function with the empirical distribution function. The
smaller the $\chi^2$, the better the expected fit of the model to the sample being tested (Haktanir, 1991).

The K-S test was used to assess the performance of individual cases as recommended in Haktanir (1991), Mamman et al. (2017), Zhang (2007). The statistic determines the distance between the estimated distribution function of the reference distribution and the empirical distribution function of the sample (Haktanir, 1991).

The MARE is the index whose value is determined between the median of the observed flows and their equivalents
calculated from the estimated distribution. This measure of model fit error is most applicable for engineering practice because it provides a quantitative estimate of high flows (Szulczewski and Jakubowski, 2018). Similar methods used in practice were also applied by Beskow et al. (2015), where on the one hand they used the KS, $\chi^2$, and on the other hand they





calculated the maximum, minimum and average Relative Absolute Error (RAE). Also Cassalho et al. (2018), used the RAE methods.

## 3 Results and discussion

The MK test showed no trends neither in the AM method (except for the O sample) nor in POT (except for samples BB and O). In most samples the p-value is below 0.05 (tab.2). Based on the test statistics, the Bu and O samples show a negative trend. Also Bezak et al. (2014) used this test for samples from three periods: 1895–2010, 1895–1952 and 1953–2010. The test indicated that all samples had a negative or positive trend. As they report, in all cases the test results were not statistically significant. Also, based on the test result, which was not statistically significant (5%) Cassalho et al. (2018) rejected 7 out of 113 series for the Rio Grande do Sul in Brazil. The MK test was used by Młyński et al. (2018) to check the observation series of 9 rivers from the Vistula River basin. They also relied on a significance level of 5%. Most samples did not meet this criterion.

Test B showed that for two samples: MPT and O analyzed in the AM method, the series are not random. Thus, in these cases the H0 hypothesis was rejected. This was assessed based on test statistics. The p-value in the AM method was above 0.05 for the samples BB, MPT, O. However, in the POT method the H0 hypothesis was not rejected, and the p-value was below p = 0.05 (tab.2). Bezak et al. (2014) used the von Neumann's ratio test whose test statistics were compared with a critical value. This test is based on a rank version proposed Bartels (1982) for testing a series for randomness.

The SNHT test rejected the H0 hypothesis for samples Bu and Wi in the POT method. This was assessed based on test statistics. In this test, the p-value was above p = 0.05 for most samples. The only exceptions are sample Wi in the AM method and samples Bu, MPSW, Wi in the POT method (tab.2). Moreover, Bezak (2014) used SNHT to assess the homogeneity of data from the Litija hydrological station on the Sava River. They determined the statistics of the test, which they compared with the critical value provided by Khaliq and Ouarda (2007). Rutkowska (2015) also used this nonparametric test for seven rivers located in the US using the methodology presented in Khaliq and Ouarda (2007).

Figure 4a shows the size of each sample after the thresholds in the POT method have been applied. These values were compared with the number of samples from the AM method. The threshold determined by the Hill plot method allowed us to obtain the largest amounts of the following samples: Bu and O. In the case of the BB, MPT, MPSW, Wi the largest amount was obtained by the AM method. Figure 4b presents the values of two analyzed thresholds in the POT method. The MAMRF method allowed us to obtain the lowest threshold value in most samples.

Analysis of the p-value K-S test, presented in Table 3, showed the need to reject the following distributions:

a) The 3W distribution in the MM estimation method, in all samples analyzed and in the methods: AM and POT.

b) The 3P3 distribution with estimation of parameters by MM and MLE methods for Wi samples in the AM and POT methods and for Bu and MPSW samples in the POT methods.





345 c) The 3W distribution with MLE estimation of parameters for the MPT sample in the AM and POT (MAMRF) methods, and in the case of the 3W distribution (MMM) and the AM method for BB.

d) Only 3LN and GEV distributions were not rejected in all samples.

Zhang (2007) reports that He studied the GPD distribution using MLE, MM, PWM, likelihood moment estimators (LMEs) estimators. He obtained a p-value close to 1 in the K-S goodness-of-fit test for each of the four estimates in the analyzed

350 distribution, which indicates that GPD distribution fits very well with empirical data.

Analysis of the p-value $\chi^2$ test, presented in Table 4, showed the need to reject the following distributions:

a) The 3P3 distribution in the MM estimation method in the POT (Hill plot) method for MPSW and O samples.

b) The 3LN (MMM) distribution in the POT (MAMRF) method for BB sample.

c) The 3W distribution in the MLE, in the AM method for the MPT sample and in the POT (Hill plot) method for the BB

355 sample.

d) The 3W distribution in the MM estimation method in the POT (Hill plot) method for the BB sample.

e) The 3W (MMM) and GEV (MLE) distributions in the POT (MAMRF) method for the BB sample.

f) The GGEV distribution in the AM method for sample BB.

According to Haktanir (1991), the statistic value of the $\chi^2$ test of a three-parameter distribution can be less than that of a two-

360 parameter distribution. Nevertheless, the statistically significant difference of the former can be worse than the latter. Similar results were obtained by Mamman et al. (2017), investigated the probability distributions of the Gum, LN and Normal models, which matched the river inflows of Kainji Reservoir in New Bussa, Niger State, Nigeria. They reported that the calculated statistics of the $\chi^2$ test was less than 1 for the Gum distribution, which indicates that this distribution is strong and there is a strong linearity between the observed and the predicted reservoir inflow. In turn, Szulczewski and Jakubowski

365 (2018) analyzed the results of the p-value of the $\chi^2$ test for P3, LN, GEV and MIX Gamma + GEV distributions with empirical values of selected rivers in the Upper Oder River Basin. They stated that for the Oder River in the Trestno and Korzeńsk profile $p < 0.05$ for two distributions and only the mixed distribution (MIX Gamma + GEV) ensures the best fit. In the case when the value of $p > 0.05$ for the analyzed distribution, then it showed the lack of the best fit of the empirical distribution with the theoretical distribution.

The p-values given in Tables 3 and 4 depend on the number of estimated parameters. The p-value determined by the K-S test (Table 3) shows that in the case of the GGEV distribution it is more difficult to work with four parameters trying to adjust this distribution. Based on the K-S test, the best-fitted parameters were obtained for the GGEV distribution for samples in the AM method: BB, MPT, Wi. In turn, the $\chi^2$ test indicates a greater number of best-fitted GGEV distribution parameters in

375 the analyzed samples and in the AM and POT methods than in the case of other distributions. In this test, other distributions obtained the best fit only for samples BB, MPSW and MPT, in the AM method, only for the samples BB, MPT in the POT Hill plot method and only for the sample BB in the case of the POT (MAMRF) method. Similar results were obtained by Szulczewski and Jakubowski (2018) for the MIX Gamma + GEV distribution in the AM method. They showed that in three samples the parameters are best fitted compared to other analyzed distributions, as can be seen from the results of the χ2 test





for the mixture distribution on nine samples. The $\chi^2$ test and K-S test was also indicated that the 3P3 distribution did not receive the best grades (tab.3,4) (with the exception of 3P3 (MLE) for the river Bu in the AM method and in the $\chi^2$ test). In the K-S test, the 3W distribution obtained the largest p-values (tab. 3) in two parameter estimation methods: MLE and MMM. This was also confirmed by the $\chi^2$ test (tab.4).

The non-rejected distributions presented in Tables 3 and 4 could be used in engineering practice. Although the K-S test

indicated rejection of more distributions and their parameter estimation methods than the $\chi^2$ test, there were still too many distributions to choose from. In this case, the MARE test is a helpful tool, which is used as the best fit test.

Table 5 presents the MARE generated by the PDFs in relation to the distributions, which obtained the lowest value. In the AM, POT (MAMRF) and POT (Hill plot) methods, the best fit was obtained by the GGEV model for all samples – (100% of analyzed cases. In turn, Escalante-Sandoval (2007) showed that the Gumbel-GEV mixed distribution generated the best

result in the AM method in 40% of the cases being analyzed – the smallest standard error (SE). Escalante-Sandoval (2007) analyzed the model at 35 gauging stations in Northwestern Mexico. As reported by Cassalho et al. (2018) multi-parameter distributions showed lower RAE errors than the 2-parameter distributions. Therefore, the multi-parameter distributions led to better results. This applies especially to the WAK and KAP distributions, which require the estimation of up to four parameters, similarly to the GGEV distribution. In turn, Beskow et al. (2015) showed that selected quantiles of the four-

parameter KAP distribution obtained the most satisfactory adjustment in accordance with the Anderson-Darling test and RAE, in contrast to other probablistic models.

A comparison of MARE results for two estimated thresholds in the POT method leads us to believe that the best fit of the GGEV distribution was achieved for samples BB, Bu, MPSW, O, Wi and O, in the POT (Hill plot) method, for one samle

MPT in the POT (MAMRF) method (tab.5).

Figures 5-7 show selected basic distributions for three methods: AM, POT (MAMRF) and POT (Hill plot) compared with the GGEV distribution.

The empirical curve from Figure 5a shows that there were two processes here rather than a single process.

As we know, water retention in a water reservoir constitutes an anthropogenic impact. This is indicated by the Bartels

randomness test (Tab. 2). Despite this, the GGEV distribution (MARE = 0.02) obtained a better fit than the 3-parameter distributions. The GEV distribution had a worse result (MARE = 0.07) (fig5b). The probability density function indicates that the GGEV model has a lower density than the GEV model (black line). In addition, unlike the GGEV distribution, the GEV distribution has a heavy tail (black line). This distribution behaves similarly when $\xi \in (0.06, 0.60)$, marked in green and blue, respectively.

Figure 6 shows the best-fitted GGEV distribution (a) which is compared with the GEV distribution in the POT method with MAMRF used to determine the threshold (b). The sample was selected based on the MARE test. The GEV distribution (MARE = 0.080) obtained the second result after GGEV (MARE = 0.009). The density graph of both distributions indicates that a better-fitted GGEV distribution has a lower density (c). The GEV distribution has a heavy tail (black line) (d). This distribution behaves similarly when $\xi \in (0.40, 1.20)$, marked in blue and green, respectively.





In Figure 7a, it seems that the GGEV distribution is the best fitted to the empirical distribution as opposed to the 3LN (MMM) distribution (7b). Comparing both PDFs (fig. 7c-7d), it should be stated that the 3LN distribution has a lower density. The GGEV distribution is characterized by a heavy tail (black, green and blue lines).

## 4 Conclusions

In this study, selected 3-parameter distributions (3P3, 3LN, 3W, GEV) were compared with the new distribution – GGEV
proposed by the Nascimento, Bourguignony and Leão team (2016). Two methods were used: the AM method, which is the most popular, and the POT method. The latter method had thresholds determined again by using two methods: the analytical method – MAMRF and a graphic method called Hill plot.

For the analyzed samples from Poland and the Czech Republic, the Upper Oder Basin, both from Poland and from the Czech Republic, and both from lowland and mountain rivers, the following conclusions were drawn:

1.    Out of the many methods used for estimating the 3-parameter distributions in accordance with the K-S and $\chi^2$ tests in both the AM and the POT methods, the best-fitted parameters were obtained by the MMM and by the MLE method for the 3W distribution. The K-S and $\chi^2$ tests used on the GGEV distribution did not always indicate their best fit, which is related to the estimation of four parameters. Similar to mixed distributions.

2.    The comparison of the two best fit tests indicates that the K-S test allowed to reject more distributions and estimation
methods tested than the $\chi^2$ test. This indicates that the K-S test is stronger than the $\chi^2$ test.

3.    For samples tested by the AM and POT (MAMRF and Hill plot) methods, GGEV turned out to be a better-fitted distribution than basic distributions, according to the MARE method.

4.    According to MARE, the GGEV distribution proved to be the best-fitted for samples with a clear anthropogenic activity such as the impact that a water reservoir has on sample's independence. This applies particularly to two methods AM
and POT (MAMRF and Hill plot).

5.    The GGEV distribution has advantages such as mixed distribution. However, unlike the mixed distribution, it does not require estimating such a large number of parameters.

Future research on the GGEV model and mixed models should include the sensitivity of the model to threshold changes in the POT method and the use of the model in the Regional Frequency Analysis.

*Data availability.* The source of data is the Institute of Meteorology and Water Management – National Research Institute. In order to share any data its consent is required. The data of the Institute of Meteorology and Water Management – National Research Institute have been processed.

*Author contributions.* LG designed the study and carried out the analysis, as well wrote the paper. All authors contributed to the introduction and discussion.

*Competing interests.* The authors declare that they have no conflict of interest.



**Acknowledgments**

The team would like to thank the Institute of Meteorology and Water Management of the National Research Institute for providing access to the data that has been used and processed. The tests was carried out at the Czech Technical University in

Prague and were supported by the University of Environmental and Life Sciences in Wrocław as part of the projects no. D210/0013/18, D210/0021/18, D010/0009/19.

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

**Appendix A**

**Abbreviations used in the article**

**Abbreviations of river names in the analyzed profiles:**

- Biała Głuchołaska, Głuchołazy profile – BB,

- Budkowiczanka, Krzywa Góra profile – Bu,

- Mała Panew, Turawa profile – MP_T,

- Mała Panew, Staniszcze Wielkie profile – MP_S,

- Osobłoga, Racławice Śląskie profile – O,

- Widawa, Zbytowa profile - Wi.

**Abbreviations of the method names used in Flood Frequency Analysis**:

- annual maximum – AM,

- peaks over a threshold – POT.

**Abbreviations of the names of nonparametric tests used to check the homogeneity of the samples:**





- The Mann-Kendall test – MK,
- Standard Normal Homogeneity Test – SNHT,
- The Bartels test for randomness – B.

**Abbreviations of probability distribution names:**

- Three parameters Pearson type III distribution – 3P3,
- Three - parameter lognormal distribution – 3LN,
- Three - Parameter Weibull distribution – 3W,
- Generalized extreme value distribution – GEV.
- Dual Gamma Generalized Extreme Value Distribution – GGEV

**Abbreviations of the method names used to estimate the parameters of the probability distributions:**

- The Method of Moments - MM
- The Modified Method of Moments Estimation - MMM
- The Maximum Likelihood Estimation - MLE
- The probability weighted moments - PWM
- The Markov chain Monte Carlo - MCMC

**Abbreviations of the best fit test names:**

- The Chi-squared Test - $\chi^2$,
- Kolmogorov-Smirnov - K-S,
- the Mean absolute relative error – MARE.



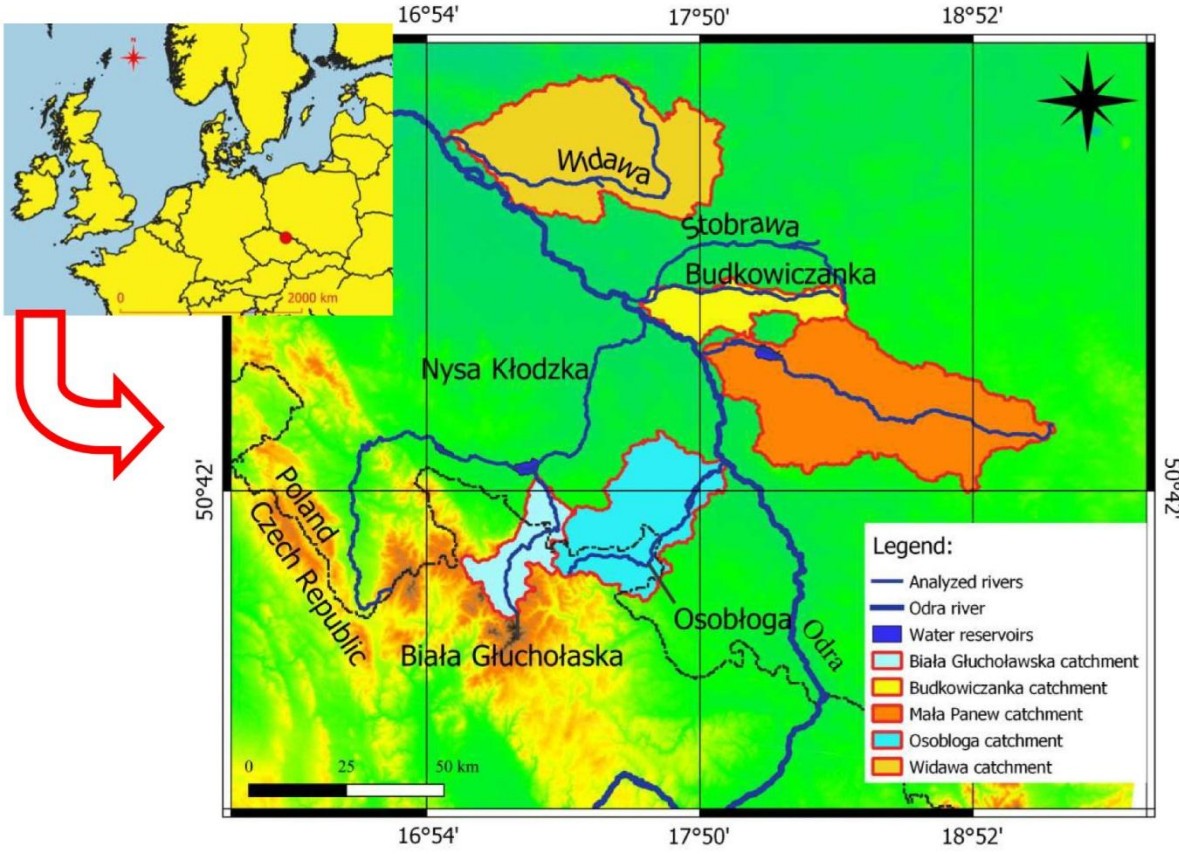

**Figure 1. Location of the analyzed tributaries of the Oder (own study based on the Hydrographic map 2019, EU-DEM 2019)**






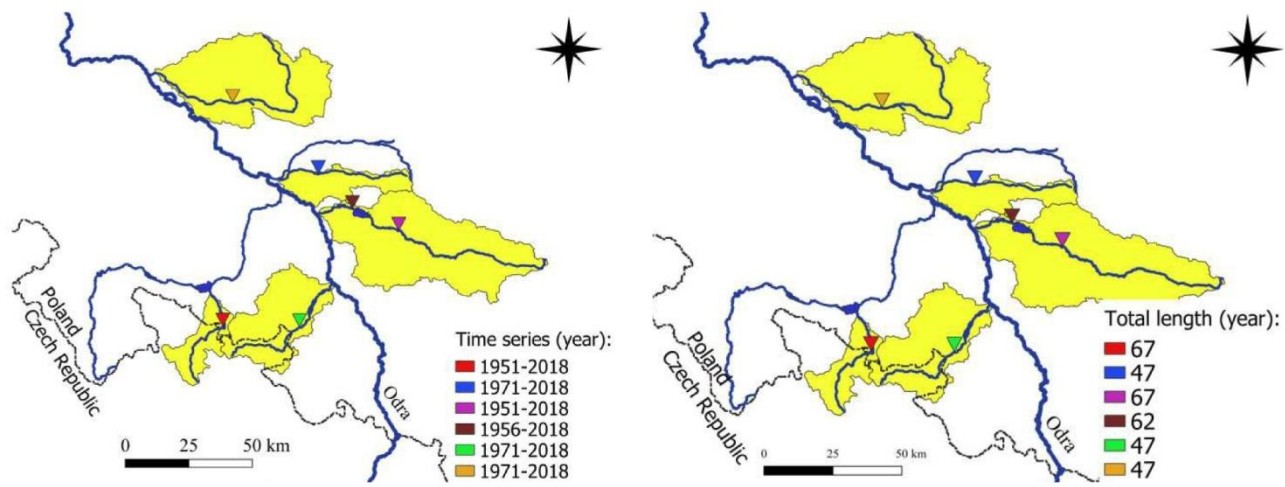

**Figure 2. Location of the water gauge stations from the analyzed profiles of the tributaries of the Oder including (a) time series of the analyzed samples and (b) their length (own study based on the Hydrographic map 2019)**





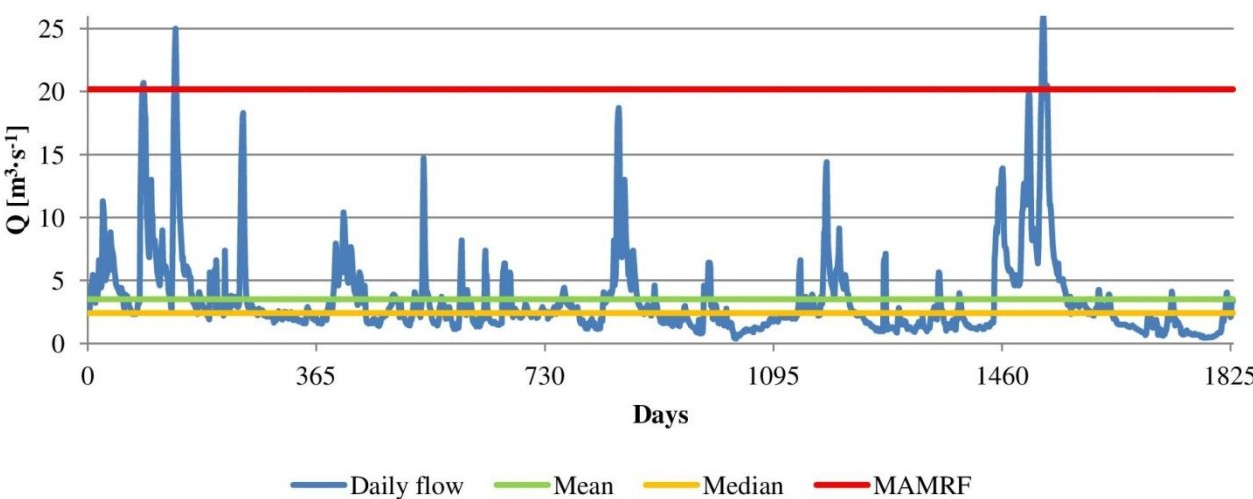

**Figure 3. Methodology of the MAMRF threshold selection over the period of five years**





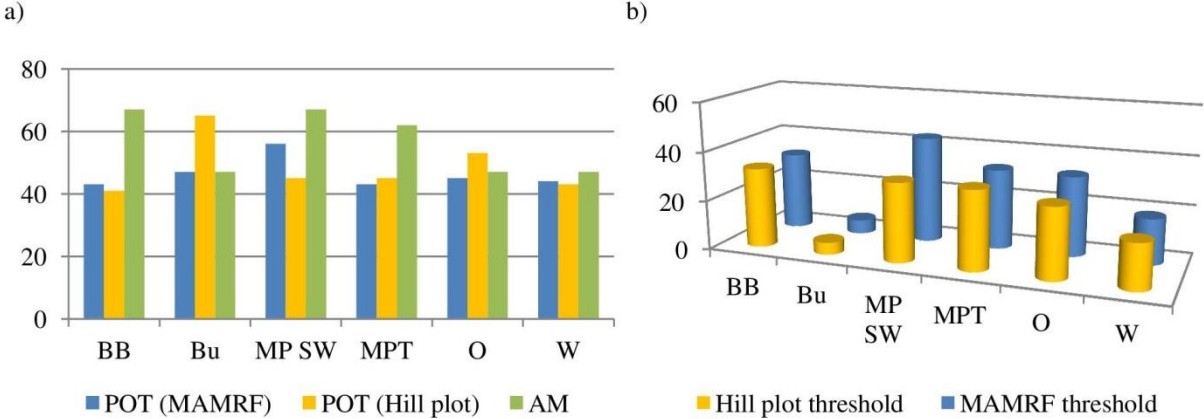

**Figure 4: Characteristics of the AM and POT samples assessing a) sample size, b) selected threshold.**








**Figure 5: Distribution curves of the AM series for the MPT sample when (a) GGEV distribution parameters are estimated with MCMC, and (b) GEV distribution parameters are obtained by using the PWM and their PDFs (c) of the GGEV (d) and GEV distribution.**


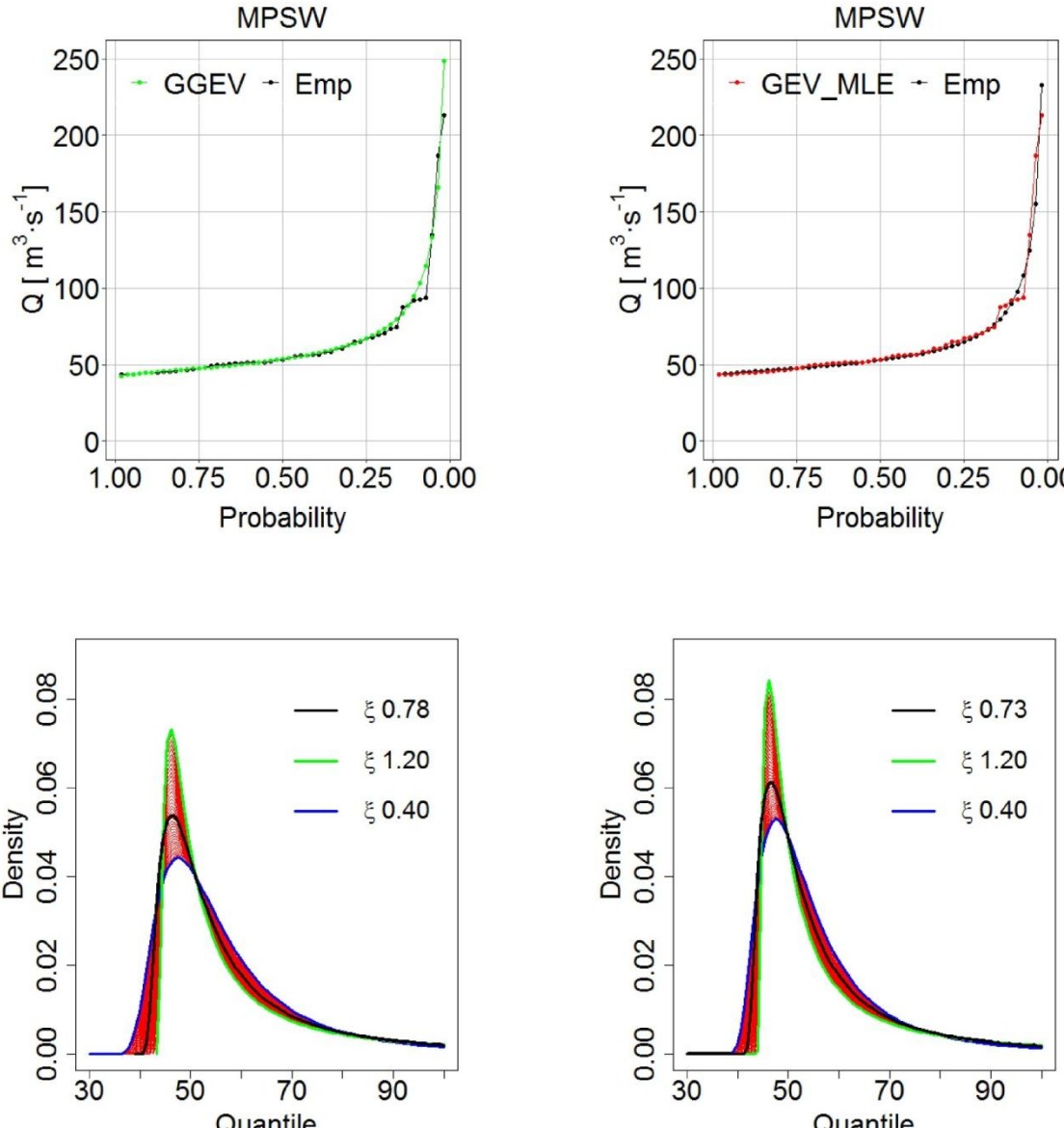

**Figure 6: Distribution curves of the AM series for the MPSW sample when (a) GGEV distribution (MCMC), and (b) GEV distribution (PWM) and their PDFs (c) of the GGEV (d) and GEV distribution.**






**Figure 7: Distribution curves of the POT-Hill plot series for sample BB when (a) the GGEV parameters are estimated with the MCMC, and (b) the 3LN distribution parameters are obtained using the MMM and their PDFs (c) of the GGEV (d) and 3LN distribution.**






**Table 1.** Profile characteristics of the river under study.

| River | Gauging station | Water gauge location | Catchment area in profile (in km$^2$) |
|---|---|---|---|
| Budkowiczanka | Krzywa Góra | 18.43 | 236.5 |
| Biała Głuchołaska (Bělá) | Głuchołazy | 18.61 | 283.0 |
| Mała Panew | Staniszcze Wielkie | 42.08 | 1107.4 |
| Mała Panew | Turawa | 17.33 | 1424.0 |
| Osobłoga | Racławice Śląskie | 27.40 | 492.0 |
| Widawa | Zbytowa | 42.77 | 720.7 |

**Source: The 2019 Report**





**Table 2.** The p-value results of the homogeneity tests.

| Sample designation | AM | | | POT | | |
|---|---|---|---|---|---|---|
| | MK | B | SNHT | MK | B | SNHT |
| **BB** | 0.040* | 0.100 | 0.500 | 0.220 rH0 | <0.001* | 0.900 |
| **Bu** | 0.030* | 0.002* | 0.200 | 0.004* | <0.001* | 0.002* rH0 |
| **MPSW** | 0.025* | 0.020* | 0.600 | 0.001* | <0.001* | 0.010* |
| **MPT** | 0.028* | 0.370 rH0 | 0.370 | 0.008* | <0.001* | 0.300 |
| **O** | 0.400 rH0 | 0.100 rH0 | 0.020* | 0.800 rH0 | <0.001* | 0.660 |
| **Wi** | 0.010* | 0.030* | 0.600 | 0.006* | <0.001* | 0.004* rH0 |

* - p-value < 0.05

rH0 - H0 hypothesis was rejected





**Table 3.** Goodness of fit of the K-S p-value for the estimated distributions.

| Samples | 3P3 (MM) | 3P3 (MLE) | 3LN (MLE) | 3LN (MM) | 3LN (MMM) | 3W (MLE) | 3W (MM) | 3W (MMM) | GEV (MLE) | GEV (PWM) | GGEV (MCMC) |
|---|---|---|---|---|---|---|---|---|---|---|---|
| **AM** | | | | | | | | | | | |
| **BB** | - | - | - | - | 0.16 | 0.73 | - | 0.59 | - | - | 0.86 |
| **Bu** | - | 0.75 | - | 1.00 | 0.10 | 1.00** | 0.005* | 1.00** | 1.00 | - | 0.52 |
| **MPSW** | - | - | 0.99 | 1.00 | 0.07 | 0.86 | - | 0.59 | 1.00 | 0.95 | 0.95 |
| **MPT** | - | - | 0.93 | 0.69 | 0.12 | 0.003* | 0.020* | 0.54 | 0.94 | 0.54 | 0.94 |
| **O** | - | 0.85 | 0.69 | 0.96 | 0.10 | 0.96 | - | 1.00** | 0.96 | 0.69 | 0.85 |
| **Wi** | <0.05* | 0.05* | 0.85 | 1.00 | 0.10 | 1.00** | - | 1.00** | 1.00 | 0.69 | 1.00 |
| **POT, MAMRF** | | | | | | | | | | | |
| **BB** | - | - | 1.00 | - | 0.86 | 0.39 | <0.05* | <0.05* | 0.86 | 0.86 | 0.32 |
| **Bu** | <0.05* | <0.05* | 0.94 | 0.94 | 0.55 | 0.84 | <0.05* | 0.99** | 0.94 | 0.55 | 0.70 |
| **MPSW** | <0.05* | - | 0.97 | 0.12 | 0.83 | 0.06 | <0.05* | 0.60 | 0.92 | 0.92 | 0.83 |
| **MPT** | 0.32 | 0.63 | 1.00 | 0.32 | 0.32 | 0.04* | <0.05* | 0.94 | 0.99 | 0.21 | 0.94 |
| **O** | 0.42 | - | 1.00 | 1.00 | 0.97 | 1.00** | <0.05* | 1.00 | 0.97 | 0.73 | 0.97 |
| **Wi** | <0.05* | <0.05* | 0.79 | 0.55 | 0.19 | 0.19 | <0.05* | 0.79** | 0.67 | 0.55 | 0.43 |
| **POT, Hill plot** | | | | | | | | | | | |
| **BB** | - | - | 0.93 | - | 0.93 | 0.11 | <0.05* | 0.93 | 0.78 | 0.99 | 0.99 |
| **Bu** | <0.05* | <0.05* | 1.00 | 1.00 | 0.95 | 1.00** | 0.01* | 1.00 | 1.00 | 0.83 | 0.70 |
| **MPSW** | - | - | 1.00 | 0.15 | 1.00 | 0.32 | 0.01* | 0.76 | 1.00 | 1.00 | 1.00 |
| **MPT** | - | - | 0.99 | 0.43 | 0.19 | 0.99 | <0.05* | 0.93 | 0.99 | 0.60 | 0.93 |
| **O** | - | - | 0.94 | 0.99 | 0.81 | 0.99** | 0.01* | 0.99** | 0.94 | 0.99 | 0.81 |
| **Wi** | 0.02* | 0.01* | 0.93 | 0.93 | 0.93 | 0.99 | 0.01* | 1.00** | 0.93 | 0.99 | 0.80 |
| | | | 8 | 6 | 1 | 5 | | 9 | 6 | 3 | 3 |

* - rejected distributions, ** - best fitted estimated distributions, confirmed by $\chi^2$ test






**Table 4.** Goodness of fit of the $\chi^2$ p-value for the estimated distributions

| Samples | 3P3 (MM) | 3P3 (MLE) | 3LN (MLE) | 3LN (MM) | 3LN (MMM) | 3W (MLE) | 3W (MM) | 3W (MMM) | GEV (MLE) | GEV (PWM) | GGEV (MCMC) |
|---|---|---|---|---|---|---|---|---|---|---|---|
| **AM** | | | | | | | | | | | |
| **BB** | - | - | 0.99 | - | 0.58 | 0.92 | - | 0.95 | - | - | <0.05* |
| **Bu** | - | 0.99 | 0.99 | 0.99 | 0.99 | 0.99** | 0.99 | 0.99** | 0.99 | - | 0.99 |
| **MPSW** | - | - | 1.00 | 0.99 | 1.00 | 0.74 | - | 0.37 | 0.99 | 0.99 | 0.99 |
| **MPT** | - | - | 0.99 | 0.99 | 0.99 | 0.04* | 0.70 | 1.00 | 0.99 | 0.99 | 0.99 |
| **O** | - | 0.96 | 0.99 | 0.99 | 0.99 | 0.99 | - | 0.99** | 0.99 | 0.84 | 1.00 |
| **Wi** | 0.16 | 0.12 | 0.99 | 0.99 | 0.99 | 0.99** | - | 0.99** | 0.99 | 0.99 | 0.99 |
| **POT, MAMRF** | | | | | | | | | | | |
| **BB** | - | - | 0.59 | - | <0.05* | - | - | 0.01* | 0.02* | 0.97 | 0.94 |
| **Bu** | 0.53 | 0.45 | 0.99 | 0.99 | 1.00 | 1.00 | 1.00 | 1.00** | 1.00 | 1.00 | 1.00 |
| **MPSW** | - | - | 0.99 | 1.00 | 1.00 | 0.83 | 0.53 | 1.00 | 1.00 | 1.00 | 1.00 |
| **MPT** | - | - | 0.99 | 0.99 | 1.00 | 1.00 | 0.97 | 1.00 | 1.00 | 1.00 | 1.00 |
| **O** | - | - | 0.99 | 0.99 | 1.00 | 1.00** | 0.98 | 1.00 | 1.00 | 1.00 | 1.00 |
| **Wi** | 0.54 | 0.43 | 0.99 | 0.99 | 1.00 | 0.99 | 1.00 | 1.00** | 1.00 | 1.00 | 1.00 |
| **POT, Hill plot** | | | | | | | | | | | |
| **BB** | - | - | 0.99 | - | 1.00 | 0.02* | 0.01* | 1.00 | 0.35 | 0.99 | 0.77 |
| **Bu** | 0.89 | 0.84 | 0.99 | 0.99 | 0.99 | 0.99 | 1.00 | 0.99 | 0.99 | 0.99 | 1.00 |
| **MPSW** | 0.01* | - | 0.99 | 0.99 | 0.99 | 0.99 | 0.99 | 0.99 | 0.99 | 0.99 | 1.00 |
| **MPT** | 0.99 | 0.99 | 0.99 | 0.99 | 0.99 | 0.06 | 0.98 | 0.99 | 0.99 | 1.00 | 0.99 |
| **O** | <0.05* | - | 0.99 | 0.99 | 0.99 | 0.99** | 0.94 | 0.99** | 0.99 | 0.99 | 0.99 |
| **Wi** | 0.35 | 0.13 | 0.99 | 0.99 | 0.99 | 0.99 | 0.99 | 0.99** | 0.99 | 0.99 | 1.00 |
| | | 1 | 8 | 7 | 13 | 9 | 6 | 13 | 12 | 11 | 10 |

\* - rejected distributions, ** - best fitted estimated distributions, confirmed by K-S test



**Table 5.** Goodness of fit of the MARE for the estimated distributions, only the best results

| Samples | AM, GGEV | POT,-MAMRF, GGEV | POT,-Hill plot, GGEV |
|---------|----------|------------------|----------------------|
| **BB** | 0.240 | 0.338 | 0.002 |
| **Bu** | 0.126 | 0.210 | 0.010 |
| **MPSW** | 0.010 | 0.009 | 0.004 |
| **MPT** | 0.020 | 0.005 | 0.011 |
| **O** | 0.100 | 0.114 | 0.006 |
| **Wi** | 0.039 | 0.002 | 0.0001 |
