# Peer review of "The application of new distribution in determining extreme hydrologic events such as floods"

_Hydrology and Earth System Sciences, 2020_

## Short Comment (SC1) · 26 Jun 2020

Konstantinos Vantas

kon.vantas@gmail.com

Dear authors,

In your paper you perform a large number of statistical tests, and some will have p values less than 0.05 purely by chance.

In my opinion you should adjust these p-values due to Multiple Comparisons, using either Bonferroni correction or any method to control the false discovery rate.

Kind regards, Dr. Konstantinos Vantas.

References (as reported in p.adjust function in {stats} package of language R):

Benjamini, Y., and Hochberg, Y. (1995). Controlling the false discovery rate: a practical

and powerful approach to multiple testing. Journal of the Royal Statistical Society Series B, 57, 289–300. http://www.jstor.org/stable/2346101.

Benjamini, Y., and Yekutieli, D. (2001). The control of the false discovery rate in multiple testing under dependency. Annals of Statistics, 29, 1165–1188. doi: 10.1214/aos/1013699998.

Holm, S. (1979). A simple sequentially rejective multiple test procedure. Scandinavian Journal of Statistics, 6, 65–70. http://www.jstor.org/stable/4615733.

Hommel, G. (1988). A stagewise rejective multiple test procedure based on a modified Bonferroni test. Biometrika, 75, 383–386. doi: 10.2307/2336190.

Hochberg, Y. (1988). A sharper Bonferroni procedure for multiple tests of significance. Biometrika, 75, 800–803. doi: 10.2307/2336325.

Shaffer, J. P. (1995). Multiple hypothesis testing. Annual Review of Psychology, 46, 561–584. doi: 10.1146/annurev.ps.46.020195.003021. (An excellent review of the area.)

Sarkar, S. (1998). Some probability inequalities for ordered MTP2 random variables: a proof of Simes conjecture. Annals of Statistics, 26, 494–504. doi: 10.1214/aos/1028144846.

Sarkar, S., and Chang, C. K. (1997). The Simes method for multiple hypothesis testing with positively dependent test statistics. Journal of the American Statistical Association, 92, 1601–1608. doi: 10.2307/2965431.

Wright, S. P. (1992). Adjusted P-values for simultaneous inference. Biometrics, 48, 1005–1013. doi: 10.2307/2532694. (Explains the adjusted P-value approach.)

---

## Author Comment (AC1) · 3 Jul 2020

Łukasz Gruss et al.

lukasz.gruss@upwr.edu.pl

Dear Konstantinos Vantas, thank you for the comment. Various authors use the Chi-square ($\chi2$) tests (Haktanir, 1991, Mamman et al. 2017, Szulczewski and Jakubowski, 2018) and the Kolmogorov-Smirnov test (K-S) (Haktanir, 1991) in similar studies. Each of these tests has been extensively used to indicate the adequacy of the distribution functions being tested. Moreover, authors such as Beskow et al. (2015) and Cassalho et al. (2018) indicated the Relative Absolute Error (RAE) test, while others like Szulczewski and Jakubowski (2018) indicated the Mean absolute relative error (MARE) test as a goodness-of-fit tests. We studied the rarely used three-parameter distributions and the new GGEV distribution proposed by Nascimento et al. (2016). We wanted to show the results obtained with the $\chi2$ test and with the K-S test separately. This is very

important because it allowed us to compare our results with those of other scientists, which we included in the discussion. Of course, we could apply Bonferroni correction, but we do not think it is really necessary. Our results obtained by the MARE test are similar to the results of Szulczewski and Jakubowski (2018) for the best-fitted test. Like others, we also used the MARE test as the most decisive to indicate the best-fitted distributions. The MARE index is a measure of the model error fit and it is most suitable for engineering practice because it provides a quantitative estimate of high flows (Szulczewski and Jakubowski, 2018). We are currently preparing a manuscript on the modeling of floods on selected European rivers, in which we will apply the Bonferroni correction You propose.

Kind regards, Łukasz Gruss (on behalf of all coauthors)

References:

Beskow S., Caldeira, T. C., Mello, C. R., and Faria, L. C.: Guedes HAS Multiparameter probability distributions for heavy rainfall modeling in extreme southern Brazil, J Hydrol: Regional Stud 4,123–133, https://doi.org/10.1016/j.ejrh.2015.06.007, 2015.

Cassalho, F., Beskow, S., de Mello, C.R., de Moura, M. M., Kerstner, L., and Ávila, L. F.: At-Site Flood Frequency Analysis Coupled with Multiparameter Probability Distributions, Water Resour. Manage., 32, 285-300, https://doi.org/10.1007/s11269-017-1810-7, 2018.

Haktanir, T.: Statistical Modelling of Annual Maximum Flows in Turkish Rivers, Hydrol. Sci. J., 36, 367–389, https://doi.org/10.1080/02626669109492520, 1991.

Mamman, M. J., Martins, O. Y., Ibrahim, J., and Shaba, M. I.: Evaluation of Best-Fit Probability Distribution Models for the Prediction of Inflows of Kainji Reservoir, Niger State, Nigeria, Air, Soil and Water Research, 10, 1–7, https://doi.org/10.1177/1178622117691034, 2017.

Nascimento F. F., and Silva W.V.M. Posterior Distribution of Extreme Value Models in R, 'MCMC4Extremes', https://cran.r-project.org/web/packages/MCMC4Extremes/index.html, 2016.

Szulczewski, W., and Jakubowski, W.: The Application of Mixture Distribution for the Estimation of Extreme Floods in Controlled Catchment Basins, Water Resour. Manage., 32, 3519–3534, https://doi.org/10.1007/s11269-018-2005-6, 2018.

---

## Referee Comment (RC1) · Anonymous Referee #1 · 25 Jul 2020

General comments The paper analyzes hydrological time series for rivers from Poland and Tchec Republic using two sampling approaches maximum annual values and peak over threshold. They fit 4 statistical distributions (univariate) with 3 parameters (3P3, 3LN, 3W, GEV) and a new distribution (extended gamma) GGEV with 4 parameters proposed by the Nascimento et al. 2016. They personal contribution is in this application to different basins from 236 to 1400 km$^2$ using R packages. They clearly indicated which R tools are used which is worth noting for readers. The paper is rich because they see how sampling methods may impact fitting results.

In my opinion the title should be reconsidered because the pioneer work is that of Nascimento et al. 2016 and was tested using hydrological data. So this distribution is not new as indicated in the title. The introduction is too large and does not focus on the

problem: application of the 4 parameter distribution using two sampling methods. The new in this paper is the use of mixed (extended) distributions. Unfortunately, the goal or the idea behind mixing is not outlined. For example, it is the case when the origin of maximum floods can be different from year (event) to year (event). So the physical meaning behind mixing is not noticed in the beginning of the paper (as in line 85). However this is the spirit of the work of Szulczewski and Jakubowski, 2018). Extended distributions would be a key word, because it was presented in this manner in the principal reference used (Nascimento et al. 2016). A section on model comparison is missed. Because authors compare 3 parameter distributions to 4 parameter, specific criteria should be adopted such as BIC and AIC.

some grammatical remarks in Abstract line 9 . without? line 103 various line 399 sample

Specific remarks : Abstract line 13 : it is not clear that authors discussed the parameter accuracy, later in this paper. Line 82 Pearson type III is 3 parameters. Its special case with 2 parameters is Gamma. Should be reformulated Line 129 the term genetic is not clear here. Why this word? authors may speak of flood generating processes Line 145 The 3 new distributions (The Dual Gamma Generalized Extreme Value Distribution (GGEV), the Exponentiated Generalized Extreme Value Distribution (EGEV)) were presented in a certain context (See Nascimento et al. 2016 "In recent years, several common distributions have been generalized via exponentiation. Let G(x) be the cdf of any continuous baseline distribution..." and Eq. 4. This context should be recalled here. Otherwise the reader who does not know the work of Nascimento et al. and other similar works about extended distributionswill not understand to general motivation of these "new" distributions lines 153 to 161 should be reformulated in order to define the objectives and the next sections of the paper line 160 why this hypothesis of the "best" ? Authors may just say that they study the adequacy of GGEV line 169 what do authors mean by profil? water level? line 172 is below meaning downstream? line 173 upstream is more adequate than below
line 200 homogeneity tests Line 225 GEV and Pareto are linked if one considers the POT model. This should be noticed somewhere because authors selected GEV (exponentiated GEVs) while using POT. In general with POT we use Pareto. line 255 why kurtosis while 3 parameters to fit? In general the smallest orders are used for distribution moments line 259 Gamma is not listed line 246. This sentence should be removed line 271 confidence level for what? do authors study the parameters confidence intervals?

line 300 "One of the goals of this article was to propose a new GGEV distribution model in the AM and POT method" this is not fully documented. line 303 what is the reference of MARE test of adequacy? line 309 it is not clear how MARE is an index. Is it MARE or an index based on MARE? line 314 A section on model comparison is missed. Because authors compare 3 parameter distributions to 4 parameter, specific criteria should be adopted such as BIC and AIC . line 318 are they significantly different from zero? If not, it is not a trend line 330 in POME application, to what extend are finding related to the level of the selected threshold? This could be more discussed. line 383 to compare fitting results of distributions involving a different number of parameters I believe that AIC or BIC criteria are more appropriate. while this is currently found in the literature, I do not believe on can rank distributions based on K-S results. K-S result is just accepting or rejecting. The value by itself has not a real meaning. One can rank distributions based of the performance of quantile estimation or parameter estimation (variance of standard error). line 410 empirical density (Kernel) should be reported in Figure 6 and figure 7 line 414 what is the reference to say that GEV distribution has a heavy tail? It is the case of Pareto, not for GEV as I know. May authors check according to El Adlouni et al. 2008 works (On the Tails of Extreme Event Distributions in Hydrology. June 2008 Journal of Hydrology 355(1):16-33)? line 430 "This indicates that the K-S test is stronger than the $\chi 2$ 430 test." this is not clear. Why is it stronger? Is thre a physical reason for rejection? line 436 point 5 . this is known from the beginning. It cannot be a conclusion Table 1 "Water gauge location Âż. what does it mean? geographic coordinates should be given source of Table 1 of what? The reference is

not correct

---

## Referee Comment (RC2) · Anonymous Referee #2 · 3 Aug 2020

Dear authors,

The paper represents a large amount of work, but I feel the presentation does not do your work justice. There are also some methodological questions that need to be clarified. The review is split into two parts: questions and comment concerning the content, and questions and comments on the style (formulations used, possible typing errors).

SUMMARY

The paper studies the applicability of several distributions for six rivers in the Upper

Oder basin. It concludes that further study is needed to determine the choice of distribution for the region.

CONTENT

General remark: it would be nice to have more justification for the use of GGEV. For instance, theoretical reasons or practical considerations such as use by one or more governments. A better fit to the data on its own is not a very strong argument. In this context the paper of Vogel and McMartin (1991) is interesting: "Probability plots for the P3 and LP3 distribution based on an estimate of the sample skew will, in general, appear more linear then they should. Essentially, the estimated sample skew acts to adjust the probability scale to make the sample, when plotted, appear more linear than it would if the the skew had been used to construct the plot." This suggests that great care must be taken to avoid overfitting and misleading fits, specially when comparing distributions with different numbers of parameters.

Line 211. Alexandersson (1986) originally intended his test to be used on series of ratios or differences with respect to a series of, possibly weighted, means of the measurements of a group of surrounding stations. Could you elaborate on how it was applied here? Given that Alexandersson (1986) assumed the ratios to be normally distributed, can you indicate why it should be suitable for series of extremes?

Line 224. To the best of my knowledge, the POT method is closely linked to extreme value theory, and the corresponding distribution to be used in fitting the data is the Generalized Pareto distribution. Please justify its use with other distributions.

Line 303. Please specify the details of the Chi-square test such as class boundaries and degrees of freedom after correction for number of fitted parameters. Please indicate how the K-S test statistic was converted to a p-value. Was the limit distribution

used? Please explicitly state how a correction was made for the number of parameters being fitted, because the standard KS test statistic distribution does not apply when comparing an empirical distribution for given data to a distribution fitted to the same data.

Line 425-438. It is customary to look not only at goodness of fit but also at the number of parameters when selecting a distribution. This is done to avoid rewarding the over-fitting of data. I feel this should be added to your analysis. Especially because in a combination of POT and GGEV there are actually five parameters being chosen.

Line 429. The purpose of both tests in your paper is not to simply reject the null hypothesis, but to reject the null hypothesis when the alternate hypothesis is true. In that case the power of the test should be examined, not the number of combinations of distribution and fitting method it rejects. The number of rejected combinations of distribution and fitting method includes type one errors. Please clarify your meaning.

STYLE

Abstract line 3: I think "with a change-point" should be "without a change-point".

Abstract line 28: "a GGEV water reservoir". What is a GGEV water reservoir?

Line 34. Is a new paragraph here necessary? It seems a continuation of the previous lines.

Line 38-45. "During ... (Pollert, 45 2006)." This seems a series of disconnected sentences, please consider rewriting.

Line 59, 60. "Therefore ...". The preceding part of this paragraph states the importance of time series analysis and the study of extremes. But in this sentence you decide to investigate rivers that are important to the water management of the Upper Oder

basin, seemingly unconnected to the preceding part of the paragraph. So why use "therefore"?

Line 70 "analyzes" should be "analyses".

Line 72. "FFA is also used to fit a probability distribution to an empirical distribution function ... ." As far as I know, flood frequency analysis is the process of studying past floods. Fitting a distribution to an empirical distribution function can be part of that process, but I do not see how a generic process can be used to do distribution fitting. Please clarify what you mean by FFA.

Line 75-80. "In time series modeling ...". Jump to a new topic (independence, trends, etc.); please improve coherence.

Line 81-109. New topic (choice of distribution); please link it to preceding material.

Line 108. It would be nice if a clear motivation for both your choice (three or more parameters) and that of many others (to parameters) was presented. Are there specific disadvantages to three parameter distributions?

Line 110-124. New topic (choice of fitting method); please add introduction linking it to this paper.

Line 125-144. New topic; please link it to preceding material.

Line 145-156. New topic; please link it to preceding material.

Line 160. "Additionally the GGEV distribution is the best suited empirical distribution irrespective of sample independence". The GGEV is not an empirical distribution. The empirical distribution is a clearly defined concept in statistics. Do you mean the GGEV fits the data best? Are you drawing a conclusion in the introduction?

[Figure]

Line 165. There is a part of a sentence missing between "The catchments of these last two rivers are" and "The Budkowiczanka River is 56.5 km long."

Line 170. "MM" in "80.04 MM m$^3$" should be "M", but even then it is not correct as ISO prefixes bind closely to the unit, so $1000000\text{m}^3 = 1\text{hm}^3$.

Line 175. Gruss et al (2019) place the source of the Widawa at 109.02 km of the river's course. How does that relate to the length of 114.6 km mentioned here?

Line 179. Sentence ends with "a Normal Pool Capacity of 1 MM cm$^3$ "; I expect this should be 1 hm$^3$ .

Line 187-193. Should most of this not be in the introduction?

Line 201. "and change point detection" should be "and the presence of change points".

Line 211. "used to analyze the change-point". Phrasing seems to assume there is a change point; do you mean: "used to check for the presence of a change point" ?

Line 247. What is meant here by "verified"?

Line 255. The term "empirical input moments" is not in use as far as I know; please write "empirical moments" instead.

Line 257. "The probability of this sample must be maximal, because the sample observed comes from many other possible samples (Haktanir, 2009)." Please either remove this sentence or replace it by a longer explanation. As it stands, it does not help the reader to understand the method.

Line 260. "In the gamma distribution developed by Becker and Klößner (2017), ... ". Becker and Klößner (2017) did not develop the Gamma distribution but a package

for the Pearson distribution system. Moreover, the Pearson III distribution has three parameters and is therefore not usually referred to as "the" Gamma distribution which traditionally has two parameters.

Line 302. "The Chi-squared Test (q2), Kolmogorov-Smirnov (K-S), and the Mean absolute relative error (MARE) tests were widely used to indicate the adequacy of the distribution functions being tested". Meaning of "widely used" in this sentence is unclear. Do you mean in the literature, in practice, in this paper?

Line 316. "The MK test showed no trends neither in the AM method (except for the O sample) nor in POT (except for samples BB and O). " This means the MK test showed trends in both methods. I assume you meant: "The MK test showed trends neither for the AM values (except for the O sample) nor for POT (except for samples BB and O). "

Line 320. "Also, based on the test result, which was not statistically significant (5%) Cassalho et al. (2018) rejected 7 out of 113 series for the Rio Grande do Sul in Brazil." Too brief, please rewrite to make meaning clearer because at the moment it can be misunderstood. Cassalho et al. (2018) state: "Based on the non-parametric Mann-Kendall test, at a significance level of 5%, only 7 out of 113 series (Fig. 2) presented significant monotonic trend, thus, they were not used for the sequence of this study." Thus, 7 series are rejected because for those series the result was statistically significant at a significance level of 5%.

Line 322. "They also relied on a significance level of 5%. Most samples did not meet this criterion." What is the criterion you refer to? In the reference 3 out of 9 series have p-values below 5%. In your sentence the criterion is: the null hypothesis of no trend is rejected at the 5% significance level. In the present context where the aim is to select series without trend, the term "criterion" might be misinterpreted. Please rewrite this line.
Line 324. "Test B showed that for two samples: MPT and O analyzed in the AM method, the series are not random. Thus, in these cases the H0 hypothesis was rejected." Please make clear what H0 is. Given the context of this paper there are 7 candidates:

a: "There is no trend"

b: "The series is random"

c: "There is no change point"

and their combinations: a and b; a and c; b and c; a,b, and c.

Line 327. "Bezak et al. (2014) used the von Neumann's ratio test whose test statistics were compared with a critical value. This test is based on a rank version proposed Bartels (1982) for testing a series for randomness." Why is this sentence here? Should it not be in Section 2.2.1 or in the introduction?

Line 348. Typo: "He" should be "he".

Line 349. "He obtained a p-value close to 1 in the K-S goodness-of-fit test for each of the four estimates in the analyzed distribution, which indicates that GPD distribution fits very well with empirical data." The p-value is not a measure of fit; it is an indication of how likely it is to get a specific test statistic value for a random sample from a given distribution. Please emphasize this somewhere in the paper.

Line 367. "In the case when the value of $p > 0.05$ for the analyzed distribution, then it showed the lack of the best fit of the empirical distribution with the theoretical distribution." If I read Table 2 in Szulczewski and Jakubowski (2018) correctly, then $p < 0.05$ leads to rejection of the hypothesis that the sample is from the given distribution; here you state the opposite. Please clarify.

Line 372. "in the case of the GGEV distribution it is more difficult to work with four parameters trying to adjust this distribution". This is a highly unusual finding; normally,

more parameters result in a better fit. Please discuss this some more.

Line 379. chi square symbol is not displayed correctly.

Line 425. "Out of the many methods used for estimating the 3-parameter distributions in accordance with ... the best-fitted parameters were obtained by the MMM and by the MLE". MM, MMM, and MLE are the only methods mentioned in the paper; the sentence mentions two of out of three, thus the phrase "Out of the many methods" seems out of place.

Table 2. What is meant by "rH0 - H0 hypothesis was rejected."? It does not seem related to the p-values in the same column.

Table 3, footnote. The K-S statistic itself is a measure of the distance between two cumulative distribution functions, but the associated p-value is not.

---

## Author Comment (AC2) · 4 Aug 2020

Łukasz Gruss et al.

lukasz.gruss@upwr.edu.pl

Dear Referee,

Thank you for your comment. We do appreciate your constructive suggestions.

Reply to general comments:

Referee: In my opinion the title should be reconsidered because the pioneer work is that of Nascimento et al. 2016 and was tested using hydrological data. So this distribution is not new as indicated in the title. Reply: Nascimento et al. (2016) investigated new distributions on the data which were monthly maxima of water levels for the Gurgueia River and the maximum precipitation in 1931-2008 for the Barcelos Station in the north of Portugal. The latter are meteorological data. Nascimento et al. (2016) did not

investigate the new distributions using data such as the maximum annual flows or daily flows. It should be noted that in flood frequency analysis flows are the most often used data. Flows were used by various researchers quoted in our article, including Abida and Ellouze (2008), Alila and Mtiraoui (2002), Bačová-Mitková and Onderka (2010), Beskow et al. (2015), Bezak et al. (2014), Cassalho et al. (2018), Escalante-Sandoval (2007), Gharib et al. (2017), Gruss et al. (2019), Gvoždíková and Müller (2017), Haktanir (1991), Holicky and Sykora (2010), Hosking et al. (1985), Kidson and Richards (2005), Kundzewicz et al. (1999), Kundzewicz et al. (2005), Lang et al. (1999), Langbein (1949), Madsen et al. (1997), Mamman et al. (2017), Rahman et al. (2015), Stojković et al. (2017), Szulczewski and Jakubowski (2018), Yadav (1998). This kind of research was carried out by Nascimento et al. (2016) for other parameters only partially. Therefore, we cannot agree with Referee's suggestion that the title should be changed. We are the first to test the new distribution using hydrological data which are flows. Therefore, we propose to maintain the current title of our article: "The application of new distribution in determining extreme hydrologic events such as floods".

Referee: The introduction is too large and does not focus on the problem: application of the 4 parameter distribution using two sampling methods. The new in this paper is the use of mixed (extended) distributions. Unfortunately, the goal or the idea behind mixing is not outlined. For example, it is the case when the origin of maximum floods can be different from year (event) to year (event). So the physical meaning behind mixing is not noticed in the beginning of the paper (as in line 85). However this is the spirit of the work of Szulczewski and Jakubowski, 2018). Extended distributions would be a key word, because it was presented in this manner in the principal reference used (Nascimento et al. 2016). A section on model comparison is missed. Because authors compare 3 parameter distributions to 4 parameter, specific criteria should be adopted such as BIC and AIC. Reply: In our opinion, our introduction does not differ substantially from standard introductions in other articles of this kind. It contains key elements that introduce our research, such as: - The importance of using maximum observed flows to calculate the exceedance probability. The introduction also justifies

the selection of water gauge stations for hydrological analysis. - Presentation of the applied Flood Frequency Analysis (FFA) methods such as the Annual Maximum (AM) and the Peaks Over Threshold (POT) and the distributions studied in the world, along with the methods of estimating distribution parameters. - Moreover, the problem of genetic heterogeneity as well as statistical heterogeneity that each researcher encounters when using long observation series to analyze the distributions was discussed. - The introduction justifies the selection of 3-parameter distributions for our research. - The results of the research on mixed distributions were presented, as well as the research on new distributions carried out by Nascimento et al. (2016). We cannot agree with the Referee that the novelty of the article is the study of mixed distributions (as claimed by the Referee). For us, the comparison the new GGEV distribution and the 3-parameter distributions is new. Similarly, the aim of our study, stated by us in the article, is different from that mentioned by the Referee. Please note that the new GGEV distribution has 4 parameters. The first three of them appear in the GEV distribution: location, scale and shape, while the fourth parameter is an additional shape parameter. As a result, these 4 parameters shape the values determined by the GGEV distribution. On the other hand, the mixed distribution consisting of two distributions, e.g. the mixture of gamma and GEV presented by Szulczewski and Jakubowski (2018), include all the parameters from these two mixture components. In this case, 6 parameters. In our opinion, the GGEV distribution studied by us is preferred than a mixed distribution consisting of two or more distributions, because the estimation of its parameters is easier. According to Otiniano et al. (2019), extensions such as the dual gamma GEV distribution (GGEV), the exponentiated GEV distribution (EGEV), the transmuted GEV (TGEV) distribution and the q-GEV could constitute a new distribution class. In what regards the Referee's comment on adding a chapter on the comparison of models using the AIC and BIC criteria, we have to say that this would mean a significant expansion of this already extensive article. But at the same time we would like to point out that our goal was to find out which distributions, in which method (AM or POT) and with which method of estimation are best fitted to empirical data. It was not our goal to compare the distributions with each other. We used the following tests: the Chi-squared ($\chi$2), the Kolmogorov-Smirnov (K-S) and the Mean Absolute Relative Error (MARE). Our choice was not accidental because we discussed our results with other authors, including Beskow et al. (2015), Cassalho et al. (2018), Escalante-Sandoval (2007), Haktanir (1991), Mamman et al. (2017), Szulczewski and Jakubowski (2018), Zhang (2007). Our methods were the same or similar to those used by these authors. In our opinion, the comparison of models using the AIC and BIC tests could be presented in a separate, follow-up article.

Our answers to specific remarks:

Referee: Abstract line 13 : it is not clear that authors discussed the parameter accuracy, later in this paper. Reply: We thank the Referee for pointing this out. We mentioned this issue in the abstract, but we should also indicate it in the introduction at the end of line 144. We hinted that in order to use a mixed distribution, many parameters should be estimated. For example, six parameters can be difficult to estimate. We mentioned that this was a disadvantage of mixed distributions, in which the mixture components are two or more distributions. Szulczewski and Jakubowski (2018) discuss the difficulties in estimating the parameters of such mixed distributions as follows: "In the case of the MIX distribution, it is much more difficult to work with the doubled number of parameters in trying to fit the mixture distribution." Also, Vaidyanathan and Lakshmi (2016) report it as follows: "However, computing time taken by the proposed method to obtain estimates is more owing to the fact that it searches the parameter space separately for each component."

Referee: Line 82 Pearson type III is 3 parameters. Its special case with 2 parameters is Gamma. Should be reformulated Reply: We would like to thank the Author of the comment for pointing this out. There is an error in line 83: instead of Pearson type III (2P3) we actually meant log-Pearson type III (LP3). In fact, we only quoted Holická and Sákora (2010) to have studied these distributions. Their conclusions show that distributions such as log-Pearson III Type and Log-Normal are appropriate for the ob-

servation series used.

Referee: Line 129 the term genetic is not clear here. Why this word? Authors may speak of flood generating processes. Reply: We thank the Referee for drawing our attention to the formulation of the problem of genetic heterogeneity. In this paragraph, we wanted to emphasize the problem of the emergence of genetic heterogeneity as well as statistical heterogeneity encountered by the researcher when using long observational series to analyze the distributions.

Referee: Line145 The 3 new distributions (The Dual Gamma Generalized Extreme Value Distribu-tion (GGEV), the Exponentiated Generalized Extreme Value Distribu-tion (EGEV)) were presented in a certain context (See Nascimento et al. 2016 "In recent years, several common distributions have been generalized via exponentiation. Let G(x) be the cdf of any continuous baseline distribution..." and Eq. 4. This context should be recalled here. Otherwise the reader who does not know the work of Nascimento et al. and other similar works about extended distributions will not understand to general motivation of these "new" distributions Reply: We indicated that Nascimento et al. (2016) investigated 3 new distributions. However, they showed that the GGEV distribution gave the best results. Therefore, we decided to test it. In the methodology, we showed the formula for the probability density function.

Referee: Lines 153 to 161 should be reformulated in order to define the objectives and the next sections of the paper Reply: We propose to keep these lines as they are. In our opinion, goals have been defined and hypotheses have been set.

Referee: Line 160 why this hypothesis of the"best"? Authors may just say that they study the adequacy of GGEV Reply: When comparing the distributions with each other, authors such as Beskow et al. (2015), Cassalho et al. (2018), Haktanir (1991), Mamman et al. (2017), Szulczewski and Jakubowski (2018) used various tests of best fit. As the tests of the best fit, we also used the Chi-squared ($\chi$2), the Kolmogorov-Smirnov (K-S) and the Mean Absolute Relative Error (MARE). This allowed us to verify our hypothesis that the GGEV distribution is the best-fitted distribution for the samples in the Upper Odra basin.

Referee: Line 169 what do authors mean by profil? water level? Reply: It is not water level. We mean the cross section of the river where the gauge station is located. The Referee is right. We should simplify it to: "The analyzed water gauge is located at km 18.43 of its course".

Referee: Line 172 is below meaning downstream? Reply: That's exactly what we meant. Thank you for pointing this out. We will rephrase it to: "One of the gauge stations (Turawa profile) is downstream of this reservoir".

Referee: Line 173upstream is more adequate than below Reply: We agree with the Referee. We will rephrase it to: "Another water gauge (Staniszcze Wielkie profile) is located upstream of the reservoir, and the distance from it to the reservoir is about 13.9 km".

Referee: Line 200 homogeneity tests Reply: Thank you for pointing this out. We agree with Referee.

Referee: Line 225 GEV and Pareto are linked if one considers the POT model. This should be noticed somewhere because authors selected GEV (exponentiated GEVs) while using POT. In general with POT we use Pareto. Reply: Thank you very much for this suggestion. We will consider this possibility in the introduction.

Referee: Line 255 why kurtosis while 3 parameters to fit? In general the smallest orders are used for distribution moments Reply: We agree with Referee. We should remove kurtosis in this sentence.

Referee: Line 259 Gamma is not listed line 246. This sentence should be removed line Reply: Line 259 should read 3P3 instead of gamma.

Referee: Line: 271 confidence level for what? Do authors study the parameters confidence intervals? Reply: This is the confidence level of the interval. We should add

it.

Referee: Line 300 "One of the goals of this article was to propose a new GGEV distribution model in the AM and POT method" this is not fully documented. Reply: In our opinion, this is documented. We showed the results of the AM and POT methods as well as the distributions that were used in both methods.

Referee: Line 303 what is the reference of MARE test of adequacy? Reply: The MARE has been approximated in the methodology in lines 309-311. We refer to the methodology presented by Szulczewski and Jakubowski (2018).

Referee: Line 309 it is not clear how MARE is an index. Is it MAREor an index based on MARE? Reply: Using the MARE, we referred to the methods proposed by Szulczewski and Jakubowski (2018). We did this test in order to compare our results with theirs as they refer to the samples from the same basin. Szulczewski and Jakubowski (2008) mention that MARE is an index: "the MARE measure of fit, an index which is very close to the engineering intuition."

Referee: Line 314 A section on model comparison is missed. Because authors compare 3 parameter distributions to 4 parameter, specific criteria should be adopted such as BIC and AIC . Reply: We presented our standpoint on these issues in Reply to comment 2 of the general comments.

Referee: Line 318 are they significantly different from zero? If not, it is not a trend Reply: In our opinion, this was presented in the results.

Referee: Line 330 in POME application, to what extend are finding related to the level of the selected threshold? This could be more discussed. Reply: The SNHT test was performed in line with the methods proposed by Bezak (2014) and Rutkowska (2015). The discussion will be expanded.

Referee: Line 383 to compare fitting results of distributions involving a different number of parameters Ibelieve that AIC or BIC criteria are more appropriate. while this is

currently found in theliterature, I do not believe on can rank distributions based on K-S results. K-S result isjust accepting or rejecting. The value by itself has not a real meaning. One can rankdistributions based of the performance of quantile estimation or parameter estimation(variance of standard error). Reply: The K-S test is recommended by Haktanir (1991), Mamman et al. (2017) and Zhang (2007). Additionally, the K-S test was also used by Beskow et al. (2015). We have been using it for several years among other tests in distribution studies. In addition, it is a test recommended by the Association of Polish Hydrologists. Moreover, Yilmaz and Çelik (2011) reported that: "An attractive feature of this test is that the distribution of the K-S test statistic itself does not depend on the underlying cumulative distribution function being tested. Another advantage is that it is an exact test (the chi-square goodness-of-fit test depends on an adequate sample size for the approximations to be valid)". Our standpoint on the issue of presenting a comparison between models using the AIC and BIC criteria is given in the reply to comment 2 of the general comments.

Referee: Line 410 empirical density (Kernel) should be reportedin Reply: Figures 5-7 show the theoretical distribution curves and, for comparison, the empirical distribution curves (Quantile function). In our opinion, this is sufficient. Below these curves, we have shown the probability density functions for different shape parameter values.

Referee: Figure 6 and figure 7 line 414 what is the reference to say that GEV distribution has a heavy tail? It is the case of Pareto, not for GEV as I know. May authors check according to El Adlouni et al. 2008 works (On the Tails of Extreme Event Distributionsin Hydrology. June 2008 Journal of Hydrology 355(1):16-33)? Reply: We must agree with the Referee. The wording was wrong. In this paper, we compared the tails of the two distributions shown in Figures 5-7. The description applies to figures 5-7. We wanted to show that the tail of the GGEV distribution is heavier than that of the LN3.

Referee: Line 430 "This indicates that the K-S test is stronger than the$\chi 2$ 430 test." this is not clear. Why is it stronger? Is thre a physical reason for rejection? Reply: Maybe

the wording used was wrong. Both the literature and our results show that the K-S test is more powerful than $\chi 2$. This is also the conclusion that emerges from our results.

Referee: Line 436 point 5 . this is known from the beginning.It cannot be a conclusion Reply: We agree with the Referee. we will delete this conclusion.

Referee: Table 1 "Water gauge location  ÌǦz. what does it mean?geographic coordinates should be given source of Table 1 of what? Reply: Figures 1 and 2 complement the table. The table and the figures have a source. If it is necessary to enter the coordinates of water gauge stations, we will provide them. The parameters in Table 1 are used in hydrology. The water gauge location is given as the kilometer of the river course.

[revised manuscript text omitted]

Yilmaz, V., Çelik, H. (2011). A statistical approach to estimate the wind speed distribution: the case of Gelibolu region. DoÄŞuş Üniversitesi Dergisi, 9 (1), pp. 122-132.

Zhang, J.: Likelihood moment estimation for the generalized pareto distribution, Aust. N. Z. J. Stat., 49,https://doi.org/10.1111/j.1467-842X.2006.00464.x, 69–77, 2007.
* * *

---

## Short Comment (SC2) · 20 Aug 2020

The article: "The application of new distribution in determining extreme hydrologic events such as floods" analyses a new 4-parameter distribution, which is the Dual Gamma Generalized Extreme Value Distribution (GGEV) and several 3-parameter distributions (Pearson type III, Log-Normal, Weibull and Generalized Extreme Value). An interesting selection of rivers located in Poland and the Czech Republic deserves a special mention. Additionally, in these countries, there are no clear guidelines on which methods (in Flood Frequency Analysis) and which probability distributions should be used. The introduction of the article shows how many publications have been written about stochastic models, but new models are still being created. The four-parameter distribution was developed by Nascimento, Bourguignony, and Leao (2016). A novelty

in this article is the analysis of the GGEV model using a long series of observations such as water flows. The creators of the GGEV model tested it only with water levels and precipitation values. Another advantage of the work is the analysis of 3-parameter distributions because there is little information about them. The authors used the often encountered tests, such as the Kolmogorov-Smirnov test and the Chisquare test, and used the Mean Absolute Relative Error test. The latter finally decided which model was best fitted. Their work shows that the best fit according to the Mean Absolute Relative Error test was obtained with the new four-parameter distribution - GGEV. The article, if published, will be used by both scientists and practitioners. I recommend this article for publication.

---

## Author Comment (AC3) · 18 Sep 2020

Dear Referee, Thank you for your comment. We do appreciate your constructive suggestions. Below I present explanations, additions and corrections.

Reply to general comments:

Referee: General remark: it would be nice to have more justification for the use of GGEV. For instance, theoretical reasons or practical considerations such as use by one or more governments.

Reply: We agree with the Referee. Below is a justification that we propose to add to the introduction: Madsen et al. (2013) created a report of flood frequency analysis where they state that both Poland and the Czech Republic have plans for further research

activities. The important information is that in the case of Poland: the annual peak cycles of Polish rivers are a mixture of summer and winter flows. A flood regime can be affected by many factors such as land cover change, canal modifications, drainage works and presumably climate change. As further reported by Madsen et al. (2013) in the Czech Republic, in the FFA analysis, a flood regime can be affected by regional precipitation of a longer duration, occurring in catchments with an area of more than 100 km2. For water bodies with a winter flood regime, snow melting should be taken into account. Additionally, it was noted in the report that, in addition to catastrophic floods, there are flash floods in mountainous areas. In both countries, an important role is attached to data compilation because in the Czech Republic and in Poland flood frequency estimation is necessary for the design of hydraulic structures, dams, urban, hydrology, flood-hazard mapping. In the report, these countries did not indicate the application of a specific likelihood distribution for these countries, hence our proposal is a new GGEV distribution.

References: Madsen, H., Lawrence, D., Lang, M., Martinkova, M., Kjeldsen, T.R.: A review of applied methods in Europe for flood-frequency analysis in a changing environment: Floodfreq COST action ES0901: European procedures for flood frequency estimation, Department of Architecture & Civil Engineering, Wallingford, U. K., 180, 2013.

Referee: A better fit to the data on its own is not a very strong argument. In this context the paper of Vogel and McMartin (1991) is interesting: "Probability plots for the P3 and LP3 distribution based on an estimate of the sample skew will, in general, appear more linear then they should. Essentially, the estimated sample skew acts to adjust the probability scale to make the sample, when plotted, appear more linear than it would if the the skew had been used to construct the plot." This suggests that great care must be taken to avoid overfitting and misleading fits, specially when comparing distributions with different numbers of parameters.

Reply: Thank you for pointing this. We propose to extend the results with the selection

of the best-fitted distribution using information criteria - Akaike Information Criterion (AIC) and Bayesian Information Criterion (BIC).

Referee: Line 211. Alexandersson (1986) originally intended his test to be used on series of ratios or differences with respect to a series of, possibly weighted, means of the measurements of a group of surrounding stations. Could you elaborate on how it was applied here? Given that Alexandersson (1986) assumed the ratios to be normally distributed, can you indicate why it should be suitable for series of extremes?

Reply: Thank you for paying attention to this. After reviewing the documentation for the test, we find that we were unable to apply this test to determine the change point detection, and we want to withdraw from it. We propose a change to line 221: "Long time series of the six profiles were checked for trend and randomness". We propose to delete the entire paragraph on SNHT on the line 239-243 and the results and discussion on the line 329-334.

Referee: Line 224. To the best of my knowledge, the POT method is closely linked to extreme value theory, and the corresponding distribution to be used in fitting the data is the Generalized Pareto distribution. Please justify its use with other distributions.

Reply: Thank you very much for this suggestion. We wanted to supplement our answer and propose to add the justification at the end of the sentence on line 75 (is below): As reported by Bezak et al. (2014) in the POT method the Exponential and generalized Pareto distributions can be used. Instead of these distributions, one can also use the LN distribution (Adamson and Zucchini, 1984, Rosbjerg, 1987), and the Weibull (Bačová-Mitková and Onderka 2010, Dimitrov 2016,) distribution functions. Also, Wong and Li (2010) use the Weibull and gamma distributions in the POT method. Likewise, Xu et al. (2019) applied 3W and GEV in POT method. The 3W distribution provides a very good estimation of short-term extreme value. They applied two assumptions: the selected peaks are Poisson distributed, and the exceedances should be approximately independent. In their study, the dispersion index is applied to select clusters and check

the Poisson character. In turn, Dimitrov (2016) used 3W in the POT method. He points out that in the POT method, all independent response peaks, which exceed a certain high threshold level, are included in the analysis. Addition, make sure that each peak corresponds to an independent event.

References: Dimitrov N.: Comparative analysis of methods for modelling the short-term probability distribution of extreme wind turbine loads, Wind Energy, 19, 717–737 10.1002/we.1861 , 2016.

Xu S., Ji C.Y., Guedes Soares C.: Estimation of short-term extreme responses of a semi-submersible moored by two hybrid mooring systems, Ocean Eng, p. 190106388, https://doi.org/10.1016/j.oceaneng.2019.106388, 2019.

Wong T.S.T., Li W.K.: A threshold approach for peaks-over-threshold modeling using maximum product of spacings. Stat Sini 20(3),1257–1272, www.jstor.org/stable/24309490, 2010.

Adamson, P.T. and Zucchini, W.: On the application of a censored log-normal distribution to partial duration series of storms. Water SA, 10 (3), 136–146, 1984.

Rosbjerg, D.: On the annual maximum distribution in dependent partial duration series. Stochastic Hydrology and Hydraulics, 1 (1), 3–16. doi:10.1007/BF01543906, 1987,

Referee: Line 303. Please specify the details of the Chi-square test such as class boundaries and degrees of freedom after correction for number of fitted parameters.

Reply: We propose to supplement the text with the following entry at the end of line 305: The smaller the $\chi 2$, the better the expected fit of the model to the sample being tested (Haktanir, 1991). In calculating the statistics, the R package 'stats' was used - 'chisq.test' function. with continuity correction performed.

References: R Core Team and contributors worldwide: The R Stats Package, 'stats', https://stat.ethz.ch/R-manual/R-devel/library/stats/html/00Index.html, 2020.

Referee: Please indicate how the K-S test statistic was converted to a p-value. Was the limit distribution used?

Reply: In the one-sample two-sided case, exact p-values are obtained as described in Marsaglia, Tsang & Wang (2003). In order to obtain p-value, the gofTest function uses an algorithm written in C. The p-value is an indication of how likely it is to get a specific test statistic value for a random sample from a given distribution. Using the gofTest functions, there was no limit to the use of the distribution. R Package 'EnvStats' was used - gofTest function that calculates statistics and p-value. We propose to supplement the text with the following entry at the end of line 308: In calculating the statistics, the R package 'stats' was used - 'chisq.test' function. with continuity correction performed.

Referee: Please explicitly state how a correction was made for the number of parameters being fitted because the standard KS test statistic distribution does not apply when comparing an empirical distribution for given data to a distribution fitted to the same data.

Reply: The K-S statistic (Dmax) was calculated using the 'gof.Test' function and a p-value was obtained for each tested distribution. The p-value was calculated only if the data follow a specified distribution.

Referee: Line 425-438. It is customary to look not only at goodness of fit but also at the number of parameters when selecting a distribution. This is done to avoid rewarding the overfitting of data. I feel this should be added to your analysis. Especially because in a combination of POT and GGEV there are actually five parameters being chosen.

Reply: In the methods, we showed which distributions have as many parameters. Afterwards, in the results in Figure 4 (a) we showed the sample sizes, and in Figure 4 (b) the threshold sizes in the POT method. We have a lot of results that we did not include in the article. Please specify more precisely what we could include?
Referee: Line 429. The purpose of both tests in your paper is not to simply reject the null hypothesis, but to reject the null hypothesis when the alternate hypothesis is true. In that case the power of the test should be examined, not the number of combinations of distribution and fitting method it rejects. The number of rejected combinations of distribution and fitting method includes type one errors. Please clarify your meaning.

Reply: We agree with the Referee. Therefore, we propose to delete the second conclusion from line 429.

STYLE

Referee: Abstract line 3: I think "with a change-point" should be "without a change-point".

Reply: We agree with the Referee. We propose to correct the sentence: However, in order to use distributions, the data must be random, without a change-point and should not have a trend.

Referee: Abstract line 28: "a GGEV water reservoir". What is a GGEV water reservoir?

Reply: We agree with the Referee. We propose to correct the sentence: This distribution turned out to be the best fit especially for the sample whose independence is affected by the presence of a water reservoir.

Referee: Line 34. Is a new paragraph here necessary? It seems a continuation of the previous lines.

Reply: We agree with the Referee.

Referee: Line 38-45. "During ... (Pollert, 45 2006)." This seems a series of disconnected sentences, please consider rewriting.

Reply: We agree with the Referee. We propose to correct the sentence: Many floods of different intensity and extent took place on the Oder and its tributaries in the 20th century and in the beginning of the 21st century (Dubicki et al., 2005). The flood that

occurred in Poland in the Oder and the Vistula basins in the summer 1997 caused 54 fatalities and material losses estimated at billions of USD (Kundzewicz et al., 1999). Afterwards extreme fluvial flooding took place in many parts of the Czech Republic in August 2002. This flood overwhelmed most of existing flood protection systems and caused damage exceeding EUR 3 billion (Holická and Sákora, 2010).

We propose to delete some of the text that begins with the sentence "During the catastrophic flood ..." on line 38 up to the end of line 45.

Referee: Line 59, 60. "Therefore ...". The preceding part of this paragraph states the importance of time series analysis and the study of extremes. But in this sentence you decide to investigate rivers that are important to the water management of the Upper Oder basin, seemingly unconnected to the preceding part of the paragraph. So why use "therefore"?

Reply: We agree with the Referee. We propose to correct the sentence: Therefore, we decided that our research would be carried out on rivers, whose proper use has a significant impact on water management and which play an important role in designing hydrotechnical structures of the Upper Oder basin.

Referee: Line 70 "analyzes" should be "analyses".

Reply: We agree with the Referee. We propose to correct the sentence: Long time observation series were processed using the Flood Frequency Analysis (FFA), so that the distribution analyses could be carried out later.

Referee: Line 72. "FFA is also used to fit a probability distribution to an empirical distribution function ... ." As far as I know, flood frequency analysis is the process of studying past floods. Fitting a distribution to an empirical distribution function can be part of that process, but I do not see how a generic process can be used to do distribution fitting. Please clarify what you mean by FFA.

Reply: The authors meant: FFA is also used to fit a probability distribution to a given

maximum flow series dataset in order to estimate the annual exceedance probability for a given flood flow. We propose to correct the sentence: Long time observation series were processed using the Flood Frequency Analysis (FFA), so that the distribution analyzes could be carried out later. FFA is often adopted to investigate the relationships between flood magnitude and the corresponding frequency of occurrence (Gharib et al., 2017). FFA is also used to fit a probability distribution to a given maximum flow series dataset in order to estimate the annual exceedance probability for a given flood discharge (Rahman et al., 2015; Haktanir, 1991; Lang et al., 1999; Silva et al., 2012; Yadav and Pande, 1998).

Referee: Line 75-80. "In time series modeling ...". Jump to a new topic (independence, trends, etc.); please improve coherence.

Reply: We agree with the Referee. We would like to give it from a new paragraph.

Referee: Line 81-109. New topic (choice of distribution); please link it to preceding material.

Reply: We agree with the Referee. We suggest adding a sentence: In turn, Barets (1982) reports that testing of sample randomness is of fundamental importance in statistics. In the estimation of the distribution of AM and POT method's, it is a generally accepted assumption that the sequence of observations is the independent and identically distributed (Gharib et al., 2017, Szulczewski and Jakubowski, 2018). Over the past 20 years, research has been conducted on testing various distributions and methods for estimating their parameters have been developed.

Referee: Line 108. It would be nice if a clear motivation for both your choice (three or more parameters) and that of many others (to parameters) was presented. Are there specific disadvantages to three-parameter distributions?

Reply: On lines from 106 to 109 is the information about 3-parameter distributions. Their disadvantage is the difficulty in estimating parameters, as reported by Kidson

and Richards (2005). We also mention this topic in the results and discussions. We suggest leaving it like this.

Referee: Line 110-124. New topic (choice of fitting method); please add introduction linking it to this paper.

Reply: We agree with the Referee. We suggest adding a sentence: The parameters of the probability distributions should be estimated. However, the estimators may not be unique in a given dataset, and thus can provide multiple solutions (Langat et al. 2019).Various methods of estimating distribution parameters have been studied. Different scientists came to different conclusions. For the LN, P3 and GEV the Maximum Likelihood Estimator (MLE) is recommended (Szulczewski and Jakubowski, 2018), whereas the L-moments method was used for the GEV, LN3, P3, GLO, KAP and WAK (Cassalho et al., 2018).

Referee: Line 125-144. New topic; please link it to preceding material.

Reply: We agree with the Referee. We propose to correct the sentence: The distribution parameters are estimated based on the maximum annual series (Cassalho et al., 2018). However, factors such as anthropogenic impact, climate change or spatial distribution of precipitation generate changes in the frequency of observed floods.

Referee: Line 145-156. New topic; please link it to preceding material.

Reply: Proponujemy dodanie krótkiego wstĄŹpu i rozpoczĄŹcie zdania od nowego akapitu: We suggest adding a short introduction and starting the sentence with a new paragraph: In turn, according to Otiniano et al. (2019), new extensions of two - and three-parameter distributions were created, which may constitute a new class of distributions.

Referee: Line 160. "Additionally the GGEV distribution is the best suited empirical distribution irrespective of sample independence". The GGEV is not an empirical distribution. The empirical distribution is a clearly defined concept in statistics. Do you

mean the GGEV fits the data best? Are you drawing a conclusion in the introduction?

Reply: Thank you for this suggestion. This is not a conclusion, but it is a second hypothesis. The sentence on line 160 should read: Additionally the GGEV distribution is the best suited to the empirical distribution irrespective of sample independence.

Referee: Line 165. There is a part of a sentence missing between "The catchments of these last two rivers are" and "The Budkowiczanka River is 56.5 km long."

Reply: We agree with the Referee. The sentence "The catchments of these last two rivers are" will be deleted.

Referee: Line 170. "MM" in "80.04 MM m3" should be "M", but even then it is not correct as ISO prefixes bind closely to the unit, so 1000000m3 = 1hm3. Reply: We agree with the Referee. We propose to correct the units in this sentence.

Referee: Line 175. Gruss et al (2019) place the source of the Widawa at 109.02 km of the river's course. How does that relate to the length of 114.6 km mentioned here?

Reply: Kilometres are correct. The total length of the watercourse is 114.6 km. The Widawa river is of the second order, therefore kilometre 0 is in the mouth of the river and grows in the opposite direction to the river. It results from the hydrographic division of Poland.

Referee: Line 179. Sentence ends with "a Normal Pool Capacity of 1 MM cm3 "; I expect this should be 1 hm3 .

Reply: We agree with the Referee. We propose to correct the units in this sentence.

Referee: Line 187-193. Should most of this not be in the introduction?

Reply: We agree with the Referee. We suggest moving the sentence starting on line 189 to the end of the sentence from line 75. According to Bačová-Mitková and Onderka (2010), Bezak et al. (2014), Gharib et al. (2017), Langbein (1949), Lang et al. (1999), Kundzewicz et al. (2005), Svensson et al. (2005) the AM method is the most common

because it samples only one extreme event per year. The POT includes all peaks above a certain flow value (the threshold) (Bezak et al., 2014; Gharib et al., 2017; Kundzewicz et al., 2005; Svensson et al., 2005).

Referee: Line 201. "and change point detection" should be "and the presence of change points". Reply: We would like to remove this phrase. We have included this in response to general comments.

Referee: Line 211. "used to analyze the change-point". Phrasing seems to assume there is a change point; do you mean: "used to check for the presence of a change point" ?

Reply: We also suggest removing this phrase. We have included this in response to general comments.

Referee: Line 247. What is meant here by "verified"?

Reply: We agree with the Referee. We wanted to write that we used the GGEV distribution. We propose to change this word: Moreover, the authors used a four-parameter distribution called Dual Gamma Generalized Extreme Value Distribution (GGEV) described by Nascimento et al. (2016).

Referee: Line 255. The term "empirical input moments" is not in use as far as I know; please write "empirical moments" instead.

Reply: We agree with the Referee. We propose to correct the sentence: The Method of Moments is based on the empirical moments such as: mean, variance, skewness and kurtosis of the sample data.

Referee: Line 257. "The probability of this sample must be maximal, because the sample observed comes from many other possible samples (Haktanir, 2009)." Please either remove this sentence or replace it by a longer explanation. As it stands, it does not help the reader to understand the method.

Reply: We agree with the Referee. We propose to delete this sentence:

Referee: Line 260. "In the gamma distribution developed by Becker and Klößner (2017), ... ". Becker and Klößner (2017) did not develop the Gamma distribution but a package for the Pearson distribution system. Moreover, the Pearson III distribution has three parameters and is therefore not usually referred to as "the" Gamma distribution which traditionally has two parameters.

Reply: We agree with the Referee. We propose to correct the sentence: In the DS Packages developed by Becker and Klößner (2017), the 3P3 distribution allows negative scale parameters to allow for negative skewness.

Referee: Line 302. "The Chi-squared Test (q2), Kolmogorov-Smirnov (K-S), and the Mean absolute relative error (MARE) tests were widely used to indicate the adequacy of the distribution functions being tested". Meaning of "widely used" in this sentence is unclear. Do you mean in the literature, in practice, in this paper?

Reply: The authors meant literature. We propose to complete the sentence:

The Kolmogorov-Smirnov (K-S) (Haktanir, 1991), the Chi-squared Test ($\chi$2) (Haktanir, 1991, Langat et al., 2019, Mamman et al, 2017, Zhang, 2007), and the Mean absolute relative error (MARE) tests were widely used to indicate the adequacy of the distribution functions being tested (Szulczewski and Jakubowski, 2018).

Referee: Line 316. "The MK test showed no trends neither in the AM method (except for the O sample) nor in POT (except for samples BB and O). " This means the MK test showed trends in both methods. I assume you meant: "The MK test showed trends neither for the AM values (except for the O sample) nor for POT (except for samples BB and O). "

Reply: We agree with the Referee. We propose to correct the sentence: The MK test showed trends neither for the AM values (except for the O sample) nor for POT (except for samples BB and O).

Interactive
comment

Referee: Line 320. "Also, based on the test result, which was not statistically significant (5%) Cassalho et al. (2018) rejected 7 out of 113 series for the Rio Grande do Sul in Brazil." Too brief, please rewrite to make meaning clearer because at the moment it can be misunderstood. Cassalho et al. (2018) state: "Based on the non-parametric Mann-Kendall test, at a significance level of 5%, only 7 out of 113 series (Fig. 2) presented significant monotonic trend, thus, they were not used for the sequence of this study." Thus, 7 series are rejected because for those series the result was statistically significant at a significance level of 5%.

Reply: We agree with the Referee. We propose to correct the sentence: Cassalho et al. (2018) report that based on the MK test, at a significance level of 5%, seven samples from 113 for the Rio Grande do Sul in Brazil presented significant monotonic trend. Thus, these seven series are rejected.

Referee: Line 322. "They also relied on a significance level of 5%. Most samples did not meet this criterion." What is the criterion you refer to? In the reference 3 out of 9 series have p-values below 5%. In your sentence the criterion is: the null hypothesis of no trend is rejected at the 5% significance level. In the present context where the aim is to select series without trend, the term "criterion" might be misinterpreted. Please rewrite this line.

Reply: We agree with the Referee. We propose to correct the sentence: Also, The MK test was used by Młyński et al. (2018) to check the significance of the trend. The study was conducted for the significance level of $\alpha$ = 5%. The values received from MK test revealed that the trends of annual peak flow, for the investigated periods, in the catchments of the Grajcarek, Wołosaty and Hoczewka streams (the three from 9 investigated streams from the Upper Vistula River basin) were significant.

Referee: Line 324. "Test B showed that for two samples: MPT and O analyzed in the AM method, the series are not random. Thus, in these cases the H0 hypothesis was rejected." Please make clear what H0 is. Given the context of this paper there are 7

candidates: a: "There is no trend" b: "The series is random" c: "There is no change point" and their combinations: a and b; a and c; b and c; a,b, and c.

Reply: In subsection 2.2.1. the null and alternative hypotheses were described for each test (MK and B) separately. I used the MK test to check the presence of a trend and the B test to check the randomness of the sample. We propose to merge the paragraphs on lines 202 and 208. The existing paragraphs may have confused the reader into thinking that the hypotheses on line 208 are for all tests in this section. In our opinion, the text should be legible after these changes. However, in the results, the MK and B tests are separated by paragraphs. we would like to remove the SNHT test as it cannot be used for this data. I wanted the tested samples to be verified in terms of trend and randomness, which should raise the level of work.

Referee: Line 327. "Bezak et al. (2014) used the von Neumann's ratio test whose test statistics were compared with a critical value. This test is based on a rank version proposed Bartels (1982) for testing a series for randomness." Why is this sentence here? Should it not be in Section 2.2.1 or in the introduction?

Reply: We agree with the Referee. We propose to remove the sentence from line 328: "This test is based on a rank version proposed Bartels (1982) for testing a series for randomness" and move it to line 217 as follows:

In the Bartels test for randomness (Bartels, 1982), (B) the null hypothesis that the sample is random is tested against the alternative hypothesis that the data is significantly different from random. This test is based on a rank version proposed Bartels (1982) for testing a series for randomness. A two-sided test was performed.

Additionally, we suggest extending the discussion of test B with the sentence on line 327:

Bezak et al. (2014) used von Neumann's ratio test whose to assess the homogeneity of data from the Litija hydrological station on the Sava River. The statistic values received

from this test were compared with critical values. The test showed that the analysed data are homogeneous for the AM series for the periods 1895–2010, 1895–1952 and 1953–2010.

Referee: Line 348. Typo: "He" should be "he".

Reply: We agree with the Referee. We propose to correct the sentence: Zhang (2007) reports that he studied the GPD distribution using MLE, MM, PWM, likelihood moment estimators (LMEs) estimators.

Referee: Line 349. "He obtained a p-value close to 1 in the K-S goodness-of-fit test for each of the four estimates in the analyzed distribution, which indicates that GPD distribution fits very well with empirical data." The p-value is not a measure of fit; it is an indication of how likely it is to get a specific test statistic value for a random sample from a given distribution. Please emphasize this somewhere in the paper.

Reply: We agree with the reviewer that p-value is an indication of how likely it is to get a specific test statistic value for a random sample from a given distribution. We will insert this comment in the subsection 2.2.1. at the end of the paragraph about K-S test.

Referee: Line 367. "In the case when the value of $p > 0.05$ for the analyzed distribution, then it showed the lack of the best fit of the empirical distribution with the theoretical distribution." If I read Table 2 in Szulczewski and Jakubowski (2018) correctly, then $p < 0.05$ leads to rejection of the hypothesis that the sample is from the given distribution; here you state the opposite. Please clarify.

Reply: We agree with the Referee. We propose to correct the discussion from line 366: They stated that the goodness-of-fit hypothesis is rejected for two distributions, for the Oder River in the Trestno (LN, p-value = 0.026; GEV, p-value = 0.005) and Korzeńsk profile (P3, p-value = 0.016, LN, p-value = 0.005) and only the mixed distribution (MIX Gamma + GEV) ensures the best fit.

The sentence: "In the case when the value of $p > 0.05$ for the analyzed distribution,

then it showed the lack of the best fit of the empirical distribution with the theoretical distribution" will be deleted.

Referee: Line 372. "in the case of the GGEV distribution it is more difficult to work with four parameters trying to adjust this distribution". This is a highly unusual finding; normally, more parameters result in a better fit. Please discuss this some more.

Reply: The parameters of the GGEV distribution are more difficult to estimate than the two-parameter or three-parameter distribution. This is discussed in articles on mixed distributions. Szulczewski and Jakubowski (2018) and Vaidyanathan and Lakshmi (2016) write about it. The number of parameters is important in the chi-square test. Haktanir (2009) states that "the chi-squared value of a three-parameter model can be less than that of a two-parameter model, the probability level of acceptance of the former can be worse than the latter. There does not exist such an" effect of the number of model parameters in the KS GOF test however ". Whereas Wilks (2011) states that if "the parameters have been estimated from the data sample, then the estimating the parameters from the same batch of data used to test the goodness of fit results in the fitted distribution parameters being" tuned "to the data sample. In practice this provision can be a limitation to the use of the KS test, since it is often the correspondence between a fitted distribution and the particular batch of data used to fit it that is of interest ". In turn, for continuous distributions the K-S test usually will be more powerful than the $\chi 2$ test and so usually will be preferred (Wilks, 2011).

References: Wilks D.S.: Statistical Methods in the Atmospheric Sciences, 3rd Edition, Academic Press, in International Geophysics, San Diego, Calif, 2011.

Referee: Line 379. chi square symbol is not displayed correctly.

Reply: We agree with the Referee.

Referee: Line 425. "Out of the many methods used for estimating the 3-parameter distributions in accordance with ... the best-fitted parameters were obtained by the

[Figure]

MMM and by the MLE". MM, MMM, and MLE are the only methods mentioned in the paper; the sentence mentions two of out of three, thus the phrase "Out of the many methods" seems out of place.

Reply: We agree with the Referee. We will correct this sentence.

Referee: Table 2. What is meant by "rH0 - H0 hypothesis was rejected."? It does not seem related to the p-values in the same column.

Reply: For the MK, B, SNHT tests (the latter we would like to remove from the table) we compared the calculated statistic with the critical value. We made a note of it. If H0 was not rejected, we showed the p-value. In table 2 we show the p-value results.

Referee: Table 3, footnote. The K-S statistic itself is a measure of the distance between two cumulative distribution functions, but the associated p-value is not.

Reply: We agree with the Referee. We propose to leave the title of Table 3 because, as suggested by Denis et al., (2018) three complementary methods are available for comparing models: p-value, by the difference between the theoretical and empirical survival functions, by the likelihood value.

References: Laurent Denis, ... David Delaux, in Reliability of High-Power Mechatronic Systems 2, 2017

---

## Author Comment (AC4) · 19 Sep 2020

Dear Referee, after our last response, we add additional corrections and information suggested by the Referee.

Referee: The introduction is too large and does not focus on the problem: application of the 4 parameter distribution using two sampling methods. The new in this paper is the use of mixed (extended) distributions. Unfortunately, the goal or the idea behind mixing is not outlined. For example, it is the case when the origin of maximum floods can be different from year (event) to year (event). So the physical meaning behind mixing is not noticed in the beginning of the paper (as in line 85). However this is the spirit of the work of Szulczewski and Jakubowski, 2018). Extended distributions would

be a key word, because it was presented in this manner in the principal reference used (Nascimento et al. 2016).

Reply: Thank you very much for your suggestion. We will shorten the text on lines 81-109 (choice from distribution) and on lines 110-124 (choice from the fitting method).

Referee: A section on model comparison is missed. Because authors compare 3 parameter distributions to 4 parameter, specific criteria should be adopted such as BIC and AIC.

Reply: After re-examining, we believe that the Referee's suggestion was justified. The AIC and BIC criteria will allow the comparison of the studied distributions. Therefore, instead of a separate chapter, we suggest adding the following changes. The full text including the revision to methods, results and discussion, and conclusion is in pdf file attached.

Referee: Abstract line 13 : it is not clear that authors discussed the parameter accuracy, later in this paper.

Reply: We propose in line 138 to add at the end of the sentence: This method searches the parameter space separately for each component. Hence, the computation time is relatively long.

We propose in line 142 to add at the end of the sentence: As reported by Szulczewski and Jakubowski (2018), it is much more difficult to estimate the doubled number of parameters in a mixed distribution.

Referee: Line 160 why this hypothesis of the"best"? Authors may just say that they study the adequacy of GGEV

Reply: We wanted to supplement our answer. We put forward two hypotheses. First hypothesis: In our study we hypothesized that the GGEV distribution is the best-fitted distribution for samples, in the Upper Oder basin, for which the flow phenomenon was caused by anthropogenic activity in the catchment. Second hypothesis: Additionally

the GGEV distribution is the best suited empirical distribution irrespective of sample independence.

Referee: Line 225 GEV and Pareto are linked if one considers the POT model. This should be noticed somewhere because authors selected GEV (exponentiated GEVs) while using POT. In general with POT we use Pareto.

Reply: Thank you very much for this suggestion. We will consider this possibility in the introduction.

We wanted to supplement our answer and propose to add the justification at the end of the sentence on line 74 (is below):

As reported by Bezak et al. (2014) in the POT method the Exponential and generalized Pareto distributions can be used. Instead of these distributions, one can also use the LN distribution (Adamson and Zucchini 1984, Rosbjerg 1987), and the Weibull (Bačová-Mitková and Onderka 2010, Dimitrov 2016,) distribution functions. Also, Wong and Li (2010) use the Weibull and gamma distributions in the POT method. Likewise, Xu et al. (2019) applied 3W and GEV in POT method. The 3W distribution provides a very good estimation of short-term extreme value. They applied two assumptions: the selected peaks are Poisson distributed, and the exceedances should be approximately independent. In their study, the dispersion index is applied to select clusters and check the Poisson character. In turn, Dimitrov (2016) used 3W in the POT method. He points out that in the POT method, all independent response peaks, which exceed a certain high threshold level, are included in the analysis. Addition, make sure that each peak corresponds to an independent event.

References: Dimitrov N.: Comparative analysis of methods for modelling the short-term probability distribution of extreme wind turbine loads J Wind Energy, 10.1002/we.1861, 2015.

Xu S., Ji C.Y., Guedes Soares C.: Estimation of short-term extreme responses of a

semi-submersible moored by two hybrid mooring systems, Ocean Eng, p. 190106388, https://doi.org/10.1016/j.oceaneng.2019.106388, 2019.

Wong T.S.T., Li W.K.: A threshold approach for peaks-over-threshold modeling using maximum product of spacings. Stat Sini 20(3):1257–1272, www.jstor.org/stable/24309490, 2010.

Adamson, P.T. and Zucchini, W., 1984. On the application of a censored log-normal distribution to partial duration series of storms. Water SA, 10 (3), 136–146.

Rosbjerg, D.: On the annual maximum distribution in dependent partial duration series. Stochastic Hydrology and Hydraulics, 1 (1), 3–16. doi:10.1007/BF01543906, 1987.

Referee: Line 259 Gamma is not listed line 246. This sentence should be removed line

Reply: We propose to correct the sentence in line 259: In the Pearson DS package developed by Becker and Klößner (2017), the 3P3 distribution allows negative scale parameters to allow for negative skewness.

Referee: Line 314 A section on model comparison is missed. Because authors compare 3 parameter distributions to 4 parameter, specific criteria should be adopted such as BIC and AIC .

Reply: We propose adding the AIC and BIC criteria to the article, which we have supplemented in response to general comments (in attach).

Referee: Line 318 are they significantly different from zero? If not, it is not a trend

Reply: We agree with the Referee, the sentence is incomplete. We propose to change the sentence from line 318 and add a second one. Based on the test statistics, the BB and O samples show a negative trend, because trend statistic (Z) is negative.

Referee: Line 330 in POME application, to what extend are finding related to the level of the selected threshold? This could be more discussed.

Reply: We propose to delete the entire paragraph on SNHT on the line 239-243 and the results and discussion on the line 329-334.

Referee: Line 383 to compare fitting results of distributions involving a different number of parameters I believe that AIC or BIC criteria are more appropriate.

Reply: Our standpoint on the issue of presenting a comparison between models using the AIC and BIC criteria is given in the reply in the general comments.

Referee: Line 430 "This indicates that the K-S test is stronger than the $\chi 2$ 430 test." this is not clear. Why is it stronger? Is thre a physical reason for rejection?

Reply: We did not calculate test power for the K-S and $\chi 2$ test. Therefore, we would like to delete this conclusion (conclusion # 2)

Please also note the supplement to this comment:
https://hess.copernicus.org/preprints/hess-2020-173/hess-2020-173-AC4-supplement.pdf
* * *
[Figure]

**Supplement:**

**Response to Anonymous Referee #1 dated** 25 July 2020, attachment:

The text copied from the article is marked in italics.

**Referee: A section on model comparison is missed. Because authors compare 3 parameter distributions to 4 parameter, specific criteria should be adopted such as BIC and AIC.**

Reply: After re-examining, we believe that the judge's suggestion was justified. The AIC and BIC criteria will allow the comparison of the studied distributions. Therefore, instead of a separate chapter, we suggest adding the following changes (marked in red).
In subsection 2.2.6:

*One of the goals of this article was to propose a new GGEV distribution model in the AM and POT method. For this reason, we checked whether this distribution or the 3-parameter distributions used in these studies provided the best fit to the empirical distribution function. The Chi-squared Test ($\chi^2$), Kolmogorov-Smirnov (K-S), and the Mean absolute relative error (MARE) tests were widely used to indicate the adequacy of the distribution* functions being tested. Akaike information criterion (AIC) and Bayesian information criterion (BIC) were used to define the best fit distribution.

*The $\chi^2$ test was used to compare the selected distribution function with the empirical distribution function. The smaller the $\chi^2$, the better the expected fit of the model to the sample being tested (Haktanir, 1991,* Langat et al., 2019*).* In calculating the statistics, the R package 'stats' was used - 'chisq.test' function. with continuity correction performed.

*The K-S test was used to assess the performance of individual cases as recommended in Haktanir (1991), Mamman et al. (2017), Zhang (2007). The statistic determines the distance between the estimated distribution function of the reference distribution and the empirical distribution function of the sample (Haktanir, 1991,* Langat *et al., 2019).* R Package 'EnvStats' was used - gofTest function that calculates statistics and p-value.

*The MARE is the index whose value is determined between the median of the observed flows and their equivalents calculated from the estimated distribution. This measure of model fit error is most applicable for engineering practice because it provides a quantitative estimate of high flows (Szulczewski and Jakubowski, 2018). Similar methods used in practice were also applied by Beskow et al. (2015), where on the one hand they used the KS, $\chi^2$, and on the other hand they calculated the maximum, minimum and average Relative Absolute Error (RAE). Also Cassalho et al. (2018), used the RAE methods.*

The AIC and BIC criteria allow to compare distributions based on estimating their likelihoods L to the same data. Both AIC and BIC require the likelihood to be maximized before it can be calculated. They are calculated as follows (Sakamoto et al., 1986):

$$AIC = 2 \cdot \ln L + 2 \cdot k \tag{6}$$
$$BIC = 2 \cdot \ln L + 2 \cdot \ln N \cdot k \tag{7}$$

where $L$ is the value of the likelihood, $N$ is the number of recorded measurements, and $k$ is the number of estimated parameters.
The lower the AIC (Bezak et al., 2014, Sakamoto et al., 1986) or the BIC value, the better the fit (Sakamoto et al., 1986).

At the end of chapter 3. Results and discussion, we suggest adding:
When analyzing the AIC and BIC values in both the AM and POT methods, the lowest values were obtained for the GGEV distribution for the six analyzed profiles (Table 6). In addition, the AIC and BIC criteria showed that lower AIC and BIC values were obtained in the POT method than in the AM method, which indicates that the GGEV distribution could be used in the POT method (Table 6).However, in the studies by Bezak et al. (2014), the lowest AIC value was obtained for the LN distribution, regardless of the methods used to estimate their parameters (the lowest AIC value for LN_MM is 187.28). The highest AIC value was obtained for the GL distribution (the highest AIC value for GL _MM is 212.27). As reported by Langat et al. (2019), Tana River research in Kenya (Gariss water gauge) three-parameter distributions: 3P3 (MLE), 3LN (MLE), GEV (MLE) reached AIC and BIC values above 1000. The lowest value was obtained by the distribution 3LN (MLE) (AIC = 1081.4, BIC = 1086.0). However, the highest values were obtained for the GEV distribution (AIC = 1083.7, BIC = 1088.4). Among the three-parameter distributions analyzed in the AM method, the lowest value of AIC = 169.62 and BIC = 175.23 was obtained by the 3W (MM) distribution for the Bu sample. However, the highest value of AIC = 4168.51, BIC = 4174.13 in the AM method was obtained for the distribution 3P3 (MM); sample O (Table 6). Among the three-parameter distributions analyzed in the POT method, the lowest value of AIC = 129.05 and BIC = 134.54 was also obtained by the 3W (MM) distribution for the Bu sample. However, the

highest value of AIC = 725.93, BIC = 731.95 in the POT method was obtained for the GEV distribution (MLE); MPSW sample (Table 6).

Table 6. Goodness-of-fit information criterion

| Samples | AIC, BIC | 3P3 (MM) | 3P3 (MLE) | 3LN (MLE) | 3LN (MM) | 3LN (MMM) | 3W (MLE) | 3W (MM) | 3W (MMM) | GEV (MLE) | GEV (PWM) | GGEV (MCMC) |
|---|---|---|---|---|---|---|---|---|---|---|---|---|
| **AM** | | | | | | | | | | | | |
| **BB** | AIC | 718.64 | 615.78 | 842.95 | 980.02 | 865.14 | 702.10 | 402.59 | 706.43 | 705.44 | 708.56 | **359.33** |
| | BIC | 725.30 | 622.44 | 849.61 | 986.68 | 871.80 | 708.75 | 409.25 | 713.09 | 712.10 | 715.22 | **374.49** |
| **Bu** | AIC | 767.27 | 338.30 | 184.63 | 413.95 | 512.36 | 250.80 | 169.62 | 251.46 | 438.52 | 441.13 | **131.63** |
| | BIC | 772.88 | 343.91 | 190.25 | 419.57 | 517.97 | 256.42 | 175.23 | 257.07 | 444.14 | 446.74 | 144.70 |
| **MPSW** | AIC | 660.87 | 550.69 | 871.79 | 897.79 | 863.54 | 655.14 | 416.27 | 658.32 | 1140.73 | 1142.65 | **324.68** |
| | BIC | 667.53 | 556.24 | 878.44 | 904.45 | 870.19 | 661.80 | 422.93 | 664.97 | 1147.39 | 1149.30 | **339.84** |
| **MPT** | AIC | 476.43 | 441.40 | 727.03 | 724.54 | 696.78 | 565.28 | 365.17 | 531.02 | 1008.10 | 1005.58 | **273.36** |
| | BIC | 482.86 | 447.83 | 733.46 | 730.97 | 703.21 | 571.71 | 371.83 | 537.44 | 1014.53 | 1012.01 | **288.15** |
| **O** | AIC | 4168.51 | 427.58 | 599.34 | 675.55 | 700.96 | 448.85 | 292.11 | 452.48 | 790.76 | 793.26 | **231.55** |
| | BIC | 4174.13 | 433.19 | 604.96 | 681.16 | 706.57 | 454.46 | 298.77 | 458.09 | 796.37 | 798.87 | **244.62** |
| **Wi** | AIC | 380.44 | 429.62 | 503.16 | 735.07 | 560.34 | 353.74 | - | 354.11 | 356.73 | 362.00 | **353.49** |
| | BIC | 386.06 | 435.23 | 508.77 | 740.68 | 565.95 | **359.36** | - | 359.73 | 362.34 | 367.62 | 370.72 |
| **POT, MAMRF** | | | | | | | | | | | | |
| **BB** | AIC | 242.77 | 290.37 | 397.79 | 558.15 | 423.29 | 444.82 | 252.56 | 378.63 | 385.79 | 389.85 | 200.90 |
| | BIC | 247.98 | 295.59 | 403.00 | 563.36 | 428.50 | 450.03 | 257.77 | 383.84 | 391.01 | 395.06 | 213.16 |
| **Bu** | AIC | 241.79 | 228.51 | 197.84 | 208.07 | 310.92 | 185.11 | 129.05 | 186.70 | 191.83 | 195.16 | 100.25 |
| | BIC | 247.00 | 233.57 | 203.33 | 213.55 | 316.40 | 190.59 | 134.54 | 192.19 | 197.31 | 200.65 | 113.07 |
| **MPSW** | AIC | 344.38 | 420.10 | 480.52 | 650.15 | 497.65 | 542.12 | 306.04 | 439.56 | 725.93 | 447.65 | 232.02 |
| | BIC | 349.13 | 425.24 | 486.54 | 656.17 | 503.67 | 548.15 | 312.06 | 445.58 | 731.95 | 453.67 | 246.01 |
| **MPT** | AIC | 339.53 | 161.49 | 321.14 | 473.79 | 352.87 | 384.83 | 192.65 | 309.20 | 310.03 | 320.74 | 159.99 |
| | BIC | 344.74 | 166.70 | 326.35 | 479.00 | 358.08 | 390.04 | 197.86 | 314.41 | 315.25 | 325.96 | 172.26 |
| **O** | AIC | 430.16 | 430.16 | 428.56 | 556.33 | 459.22 | 405.00 | 239.49 | 356.14 | 363.68 | 365.86 | 187.20 |
| | BIC | 435.51 | 435.51 | 433.91 | 561.69 | 464.58 | 410.36 | 244.84 | 361.49 | 369.03 | 371.22 | 199.75 |
| **Wi** | AIC | 265.40 | 268.96 | 304.59 | 453.09 | 338.93 | 255.15 | 176.31 | 256.86 | 261.80 | 263.35 | 139.31 |
| | BIC | 270.69 | 274.24 | 309.88 | 458.37 | 344.22 | 260.43 | 181.59 | 262.15 | 267.09 | 268.63 | 151.85 |
| **POT, Hill plot** | | | | | | | | | | | | |
| **BB** | AIC | 350.27 | 362.04 | 489.83 | 534.46 | 492.80 | 435.63 | 472.33 | 480.18 | 410.30 | 473.71 | 207.74 |
| | BIC | 355.41 | 367.18 | 494.97 | 539.61 | 497.94 | 440.77 | 477.47 | 485.32 | 415.44 | 478.85 | 220.01 |
| **Bu** | AIC | 615.77 | 320.96 | 319.39 | 430.20 | 259.95 | 157.45 | 178.87 | 252.97 | 262.34 | 278.72 | 133.58 |
| | BIC | 622.25 | 327.43 | 325.86 | 436.67 | 266.43 | 163.93 | 185.34 | 259.45 | 268.81 | 285.20 | 148.37 |
| **MPSW** | AIC | 336.75 | 377.41 | 595.26 | 661.57 | 603.67 | 542.12 | 499.00 | 530.15 | 566.49 | 514.97 | 232.77 |
| | BIC | 342.77 | 383.43 | 601.28 | 667.59 | 609.69 | 548.15 | 505.02 | 536.18 | 572.51 | 521.00 | 246.76 |
| **MPT** | AIC | 274.31 | 291.74 | 325.29 | 492.44 | 357.63 | 401.97 | 192.65 | 316.63 | 322.40 | 345.74 | 167.09 |
| | BIC | 279.66 | 297.09 | 330.65 | 497.79 | 362.98 | 407.32 | 197.86 | 321.98 | 327.75 | 351.09 | 179.64 |
| **O** | AIC | 406.21 | 454.74 | 557.06 | 665.61 | 570.45 | 428.06 | 239.49 | 429.67 | 512.03 | 442.63 | 223.46 |
| | BIC | 412.06 | 460.59 | 562.91 | 671.46 | 576.30 | 433.92 | 244.84 | 435.52 | 517.88 | 448.48 | 237.01 |
| **Wi** | AIC | 319.70 | 325.44 | 386.14 | 493.38 | 407.10 | 358.19 | 341.25 | 329.46 | 349.34 | 322.56 | 138.19 |
| | BIC | 324.99 | 330.73 | 391.43 | 498.66 | 412.38 | 363.47 | 346.54 | 334.75 | 354.62 | 327.85 | 150.73 |

Additionally, in Chapter 4, Conclusions, we propose to add:

4. *According to MARE* and AIC and BIC, *the GGEV distribution proved to be the best-fitted for samples with a clear anthropogenic activity such as the impact that a water reservoir has on sample's independence. This applies particularly to two methods AM and POT (MAMRF and Hill plot).*

References:
Sakamoto, Y., Ishiguro, M., and Kitagawa G. (1986). *Akaike Information Criterion Statistics*. D. Reidel Publishing Company.

Langat P.K., Kumar L., Koech R.: Identification of the most suit-able probability distribution models for maximum, minimum, and mean streamflow. Water 11:734, 2019. https ://doi.org/10.3390/w11040734

R Core Team and contributors worldwide: The R Stats Package, 'stats', https://stat.ethz.ch/R-manual/R-devel/library/stats/html/00Index.html, 2020.